# PLK1 inhibition delays mitotic entry revealing changes to the phosphoproteome of mammalian cells early in division

Monica Gobran [1], Antonio Z Politi [1], Luisa Welp[2,3], Jasmin Jakobi[1], Henning Urlaub[2,3] & Peter Lenart [1]✉

## Abstract

**Polo-like kinase 1 (PLK1) is a conserved regulator of cell division. During mitotic prophase, PLK1 contributes to the activation of the cyclin-dependent kinase 1 (CDK1). However, the exact functions of PLK1 in prophase remain incompletely understood. Here, we show that PLK1 inhibition in synchronous G2 cell populations of multiple mammalian cell lines delays or prevents mitotic entry with high variability between individual cells. Using a mathematical model, we recapitulate this phenomenon and provide an explanation for the observed phenotypic variability. We show that PLK1-inhibited cells are delayed in a prophase-like state with low CDK1 activity that increases slowly and gradually over hours. These cells display progressively condensing chromosomes, increased microtubule dynamics, and reorganization of the actin cortex, while the nuclear envelope remains intact. We characterize this state further by phosphoproteomics, revealing phosphorylation of regulators of chromatin organization and the cytoskeleton consistent with the cellular phenotypes. Together, our results indicate that PLK1 inhibition stabilizes cells in a prophase-like state with low CDK1 activity displaying a specific set of early mitotic phosphorylation events.**

**Keywords** Cell Cycle; Mitosis; Mitotic Entry; Phosphoproteomics; Polo-like Kinase 1
**Subject Categories** Cell Adhesion, Polarity & Cytoskeleton; Cell Cycle; Computational Biology

## Introduction

While prophase is unquestionably the first phase of cell division, 'mitotic entry' is conventionally defined as the next step; the transition from prophase to prometaphase. The reason behind this definition is that the transition to prometaphase is marked by nuclear envelope breakdown (NEBD), a sudden event that is easily visible even through a simple microscope. At the molecular level, NEBD coincides with the full activation of cdk1-cyclin B, causing a massive phosphorylation of mitotic substrates, and marking the irreversible commitment to division (Gavet and Pines, 2010; Crncec and Hochegger, 2019). Cells can be easily synchronized in prometaphase by spindle poisons, which additionally enabled the detailed characterization of this state by various methods, including phosphoproteomics (Daub et al, 2008; Dephoure et al, 2008). By contrast, the onset, the exact timing, the specific cellular events and the corresponding molecular changes that occur during the preceding prophase state remain poorly characterized, to a large part due to the lack of protocols for synchronization of cells at this state.

What we do know is that after completion of DNA replication (S-phase), somatic cells enter the second gap phase (G2) to prepare for division. After spending a few hours in G2, cells proceed to prophase. In prophase, chromosomes start to condense, primarily driven by the nuclear condensin II complex (Hirota et al, 2004). The cytoskeleton also starts to reorganize: centrosomes mature, meaning that they recruit additional pericentriolar material (PCM) to increase their microtubule nucleation capacity, and subsequently separate (Lee and Rhee, 2011; Tanenbaum and Medema, 2010). Concomitantly, microtubules begin to change from the interphasic to mitotic dynamics (Ferenz and Wadsworth, 2007). Similarly, the actin cytoskeleton starts to remodel preparing the cell for rounding up. Therefore, stress fibers and focal adhesions disassemble (Chen et al, 2022), and through the activation of the RhoA pathway, cortical contractility is increased (Taubenberger et al, 2020). Furthermore, the disassembly of the nuclear envelope (NE) also begins in prophase with the release of a subset of nuclear envelope proteins (Velez-Aguilera et al, 2020). Due to phosphorylation of proteins of the nuclear pore complex (NPC), nucleo-cytoplasmic transport is altered (Linder et al, 2017; Nkoula et al, 2023). However, the overall nucleo-cytoplasmic compartmentalization remains intact during prophase.

The above changes in cellular architecture are driven by mitotic kinases, prominently Aurora A and B, polo-like kinase 1 (plk1) and cdk1. For example, in prophase Aurora B phosphorylates histone

[1]Research Group Cytoskeletal Dynamics in Oocytes, Max Planck Institute for Multidisciplinary Sciences, 11 Am Fassberg, 37077 Göttingen, Germany. [2]Bioanalytical Mass Spectrometry, Max Planck Institute for Multidisciplinary Sciences, 11 Am Fassberg, 37077 Göttingen, Germany. [3]Bioanalytics, Institute of Clinical Chemistry, University Medical Center Göttingen, 40 Robert Koch Strasse, 37075 Göttingen, Germany. ✉E-mail: peter.lenart@mpinat.mpg.de

H3, an early mark of chromosome condensation (Hsu et al, 2000). Plk1 mediates centrosome maturation by phosphorylating the PCM component pericentrin (Lee and Rhee, 2011). Plk1 also contributes to the remodeling of the nuclear envelope through phosphorylation of the nucleoporins Nup98 and Nup53 (Linder et al, 2017; Laurell et al, 2011).

In addition, Aurora kinases and plk1 are key components of the signaling network leading to activation of cdk1. In mammalian somatic cells, cdk1-cyclin B complexes are present and ready to be activated already as cells enter G2. However, cdk1 is kept inactive by phosphorylations on Thr14 and Tyr15 by wee1/myt1 kinases (Crncec and Hochegger, 2019). Cdk1 activity then begins to increase gradually in late G2, in a process that remains incompletely understood. A critical event for activation of cdk1 is the dephosphorylation of the Thr14/Tyr15 sites by cdc25 phosphatases. This is a highly regulated process receiving inputs from multiple signaling pathways (Liu et al, 2020). A major route leading to activation of cdc25 begins by cdk1/2-cyclin A phosphorylating the protein Bora, leading to the activation of Aurora A, which in turn phosphorylates and activates plk1. Plk1 then phosphorylates and activates cdc25C (Gheghiani et al, 2017). Multiple feedback loops render activation of cdk1 a bistable, switch-like process (Rata et al, 2018), in particular, cdk1 inhibits wee1/myt1 and activates cdc25 forming two positive feedback loops. Additional feedback loops exist between cdk1 and PP2A-B55, the main phosphatase responsible for dephosphorylating cdk1 substrates. A mathematical model of this signaling network was able to quantitatively recapitulate many aspects of the experimentally observed cdk1 activation kinetics (Rata et al, 2018).

Considering the above, it is clear that the cell begins to reorganize its architecture in prophase at a time when cdk1 and other mitotic kinases are already, at least partially, active. However, we lack a systematic overview of the molecular changes, in particular changes in phosphorylation states, that occur at this first stage of cell division. Previous approaches using immunolabeling of cells and subsequent FACS sorting, provided rather limited cell numbers of such transient stages as prophase (Ly et al, 2017; Kelly et al, 2021). Another study used highly synchronous cell populations released into mitosis, which relied on very tight 5-min sampling of cells and focused exclusively on chromatin associated proteins (Samejima et al, 2022). Thus, the inability to obtain a large and homogeneous prophase cell population remains a strong limitation that prevented a systematic characterization of early steps of division by phosphoproteomics so far.

Here, we investigated the effects of plk1 inhibition to gain insights into cellular events of prophase and their regulation. Previous studies using various cell lines and different methods for inhibiting plk1 showed varying effects ranging from a few hours of delay to complete inhibition of mitotic entry (e.g., Lénárt et al, 2007; Gheghiani et al, 2017). We confirm these observations and show that a high variability exists at the level of individual cells even within population of a cell line. A computational model of the signaling network recapitulates and explains this seemingly variable phenotype. By using a FRET-based probe, we show that cdk1 activity increases slowly over many hours in plk1-inhibited cells preceding NEBD. We show that these cells are delayed in a prolonged, prophase-like state, displaying progressively condensing chromosomes and increased dynamics of the actin and microtubule cytoskeleton, while the nuclear envelope remains intact. We

characterized this state further by phosphoproteomics, revealing phosphorylation of proteins involved in chromatin organization and cytoskeletal regulation, well consistent with cellular phenotypes. Together, we show that plk1 inhibition slows cdk1 activation at mitotic entry and thereby stabilizes cells in a prophase-like state, providing insights into phosphorylation events that occur early in cell division.

# Results

## Plk1 inhibition delays mitotic entry to a highly variable extent

Previous publications reported somewhat inconsistent effects of plk1 inhibition (Lénárt et al, 2007; Gheghiani et al, 2017), therefore we first wanted to compare side-by-side three cell lines that are most commonly used in cell cycle studies: two cancer cell lines, HeLa and U2-OS, and an hTERT immortalized non-cancer cell line, hTERT-RPE1. We synchronized populations of cells using S-phase arrest and treated them upon release. We then filmed the cells stained with the red fluorescent Hoechst derivatives 5'-SiR- or 5'-CP-580-Hoechst (Bucevičius et al, 2019) for 24 h (Fig. 1A). Mitotic entry (i.e., NEBD) was identified by the collapse of the condensed chromosomes (Fig. 1B). Imaging per se had no significant effect on the cells, as DMSO-treated controls progressed though division with normal timing and morphology and consistent with growth rates in culture (Fig. 1B,C).

We treated cells with increasing concentrations of BI 2536, a well-established and highly specific inhibitor of plk1 (Lénárt et al, 2007; Steegmaier et al, 2007). Starting from low concentrations expected to only partially inhibit plk1, we escalated doses to multiples of the concentrations shown to suffice for complete inhibition (100 nM for HeLa and U2-OS, 250 nM for hTERT-RPE1 cells (Steegmaier et al, 2007; Petronczki et al, 2007)). As also shown in these and other studies, the fully penetrant *polo* phenotype (chromosomes arranged in a ring shape) is a good indicator of complete plk1 inhibition, which we observed above 25, 100, and 250 nM in HeLa, U2-OS and hTERT-RPE1 cells, respectively (Fig. 1B–D). Importantly, we observed the *polo* phenotype to the same penetrance at late time points, arguing against degradation or inactivation of BI 2536 over the 24 h of imaging and consistent with the pharmacokinetic data (Steegmaier et al, 2007).

In all cell lines investigated, plk1 inhibition caused a delay in mitotic entry together with an increasing fraction of cells failing to enter mitosis. The extent of the delay and the fraction of the cells that failed to enter mitosis within the 24-h imaging period differed substantially between cell lines. In HeLa cells, plk1 inhibition caused an entry delay of 4–5 h and even at 1 µM (10-times above the full inhibitory dose) two-thirds of the cells entered mitosis. U2-OS cells showed a more extended delay of more than 12 h and only ~10% of the cells still entered mitosis at the highest 1 µM concentration. Mitotic entry in hTERT-RPE1 cells was delayed similarly to HeLa, by approximately 4–6 h, but the proportion of cells entering mitosis was substantially lower, only 8% at higher concentrations (Fig. 1C,D).

Using HeLa cells expressing PCNA-EGFP, we confirmed that the duration of S-phase was unaffected by BI 2536, indicating that the delay occurs in G2/M (Fig. EV1A,B). We could also confirm

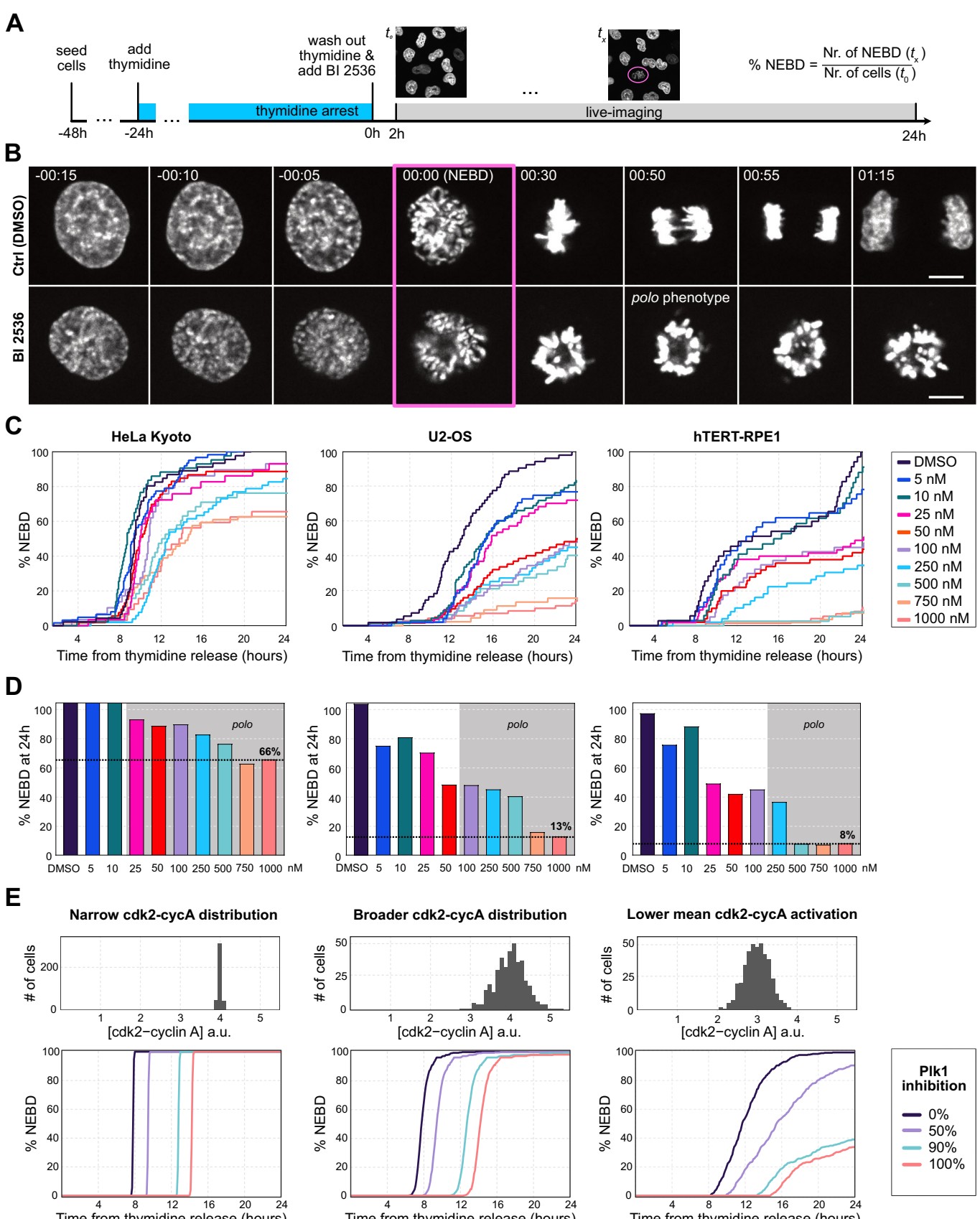

◄ **Figure 1. Plk1 inhibition causes delayed mitotic entry with highly variable timing.**

(A) Schematics of the experimental workflow. Images shown in panels below are all cropped and selected frames from such experiments. (B) HeLa cells stained with 5-SiR-Hoechst at different stages of mitosis with and without treatment with 250 nM BI 2536. Time is relative to NEBD. Scale bars are 10 μm. (C) Cumulative plots of NEBD timing in response to increasing concentrations of BI 2536 in HeLa Kyoto, U2-OS, and hTERT-RPE1 cells. (D) Bar charts show the percentage of cells that had undergone NEBD by the end of the experiment. Gray areas indicate the concentrations at which the *polo* phenotype appears. For each treatment, a total of 30 to 70 cells were analyzed. (E) Simulated NEBD timing upon plk1 inhibition using a mathematical model. Upper panels show three distributions cdk2-cyclin A levels used to simulate single cells. Lower panels show the resulting NEBD timing. Source data are available online for this figure.

that synchronization had no effect on the delay caused by BI 2536 treatment (Fig. EV1C). Furthermore, γ-H2AX staining showed no significant difference in the number of DNA damage foci, indicating that plk1 inhibition is unlikely to cause this delay by affecting DNA damage (Fig. EV1D), consistent with previous results (Gheghiani et al, 2017). To further substantiate our findings, we also tested the comparably selective and potent, but structurally unrelated plk1 inhibitor, GSK 461364 (Gilmartin et al, 2009). HeLa cells showed a very similar delay in mitotic entry as observed with BI 2536, and displayed a fully penetrant *polo* phenotype after NEBD (Fig. EV3A,B).

Taken together, our systematic survey across concentrations and cell lines confirms that plk1 activity is required for timely mitotic entry, i.e., NEBD. This effect is variable between cell lines, but we observe an even higher variability at the level of individual cells in a population. Within a single field of view, we observed cells that enter mitosis almost without delay, other cells that are delayed for many hours or that do not enter at all during the observation period. However, in all cell lines investigated, a substantial proportion of cells was able to enter mitosis even at the highest inhibitor concentrations, indicating that mitotic entry, even if delayed, can occur in the absence of plk1 activity.

## A computational model explains the variable delay in mitotic entry

To better understand the role of plk1-inhibition on mitotic entry we performed computer simulations using a previously published mathematical model of the signaling network regulating mitotic entry (Rata et al, 2018) (Fig. EV1E). However, while plk1's position and function in this signaling network has been mapped experimentally (Gheghiani et al, 2017), this was not yet included in the model. Therefore, we introduced plk1 as an activator of cdc25C downstream of cycA-cdk2 (Fig. EV1E). We then performed a Western blot for cdc25C to experimentally validate the change we introduced. This shows that the active, phosphorylated form cdc25C is absent in BI 2536-treated cells (Fig. EV1F), confirming previous reports (Gheghiani et al, 2017), and demonstrating that plk1 acts on the cdc25C branch of the mitotic entry network. Additionally, we adapted the parameter values to qualitatively match the observed kinetics of cell cycle progression in the cell lines used in our experiments (for details, see Material and Methods).

We simulated entry into G2 by a step change of cdk2-cyclin A activity from 0 to a positive value, and ran simulations at 0%, 50%, 90%, and 100% plk1 inhibition to recapitulate our BI2536 dose-response experiments. In experiments, cells do not undergo NEBD at the same time due to imperfect synchrony and cell to cell variability. We modeled this variability by slightly varying the cdk2-cyclin A activity. We show results for 3 different distributions of

cdk2-cyclin A levels (Fig. 1E). For each scenario, we computed the fraction of cells (total of 400 cells) that reached the NEBD threshold at different times (in a similar manner as we represent data in Fig. 1C).

The model captures key features of the behavior observed in experiments: for all scenarios, plk1 inhibition causes a dose-dependent delay in NEBD. For a narrow distribution of parameter values, as all cells enter mitosis at the same time, plk1 inhibition merely shifts the entry time (Fig. 1E, left panel). However, a broader distribution of cdk2-cyclin A activities, simultaneously to the delay, also lead to a broadening of mitotic entry times (Fig. 1E, middle panel). This behavior is very reminiscent of observations in HeLa cells. If the cdk2-cyclin A activity is lowered, the delay and broadening of entry times is more pronounced and a large fraction of the cells do not enter mitosis within 24 h at complete plk1 inhibition (Fig. 1E right panel). Thereby, this scenario closely recapitulates the behavior seen in U2-OS and hTERT-RPE1 cells.

Analysis of the simulations reveals that inhibition of plk1 does not change the overall, bistable behavior of the network (Fig. EV1G). Plk1 inhibition rather affects the threshold of cdk2-cyclin A activity above which there exists only the high cdk1-cyclin B steady state (i.e., saddle-node points, SN, Fig. EV1G). The closer the cdk2-cyclin A activity gets to its critical threshold, the longer the delay before nuclear envelope breakdown (NEBD). In other words, as plk1 inhibition impairs cdk1-cyclin B-independent cdc25 activation, it takes longer to reach a high level of cdc25 phosphorylation and active cdk1-cyclin B sufficient to sustain positive feedback.

Simulations of the mitotic entry network thus recapitulate the effect of plk1 inhibition on different cell lines. Importantly, the simulations show that small variabilities between individual cells and cell types can result in different outcomes ranging from a slight delay to complete block of mitotic entry. Thus, these seemingly diverse phenotypes that were interpreted as controversies in the literature and were suspected to stem from technical issues, are in fact readily explained by the inherent properties of the signaling network.

## Preceding NEBD, cdk1 activity increases slowly over hours in plk1-inhibited cells

Next, we wanted to further characterize how cdk1-cyclin B activity changes during the extended delay caused by plk1 inhibition. Therefore, we established a stable HeLa cell line expressing an optimized, Förster Resonance Energy Transfer (FRET)-based cdk activity biosensor, Eevee-spCdk (Sugiyama et al, 2024). In control cells, the sensor showed no change during G2 phase, followed by an increase in FRET starting ~90 min before NEBD (Fig. 2A,C, for single cell traces see EV2A). This increase in FRET was completely

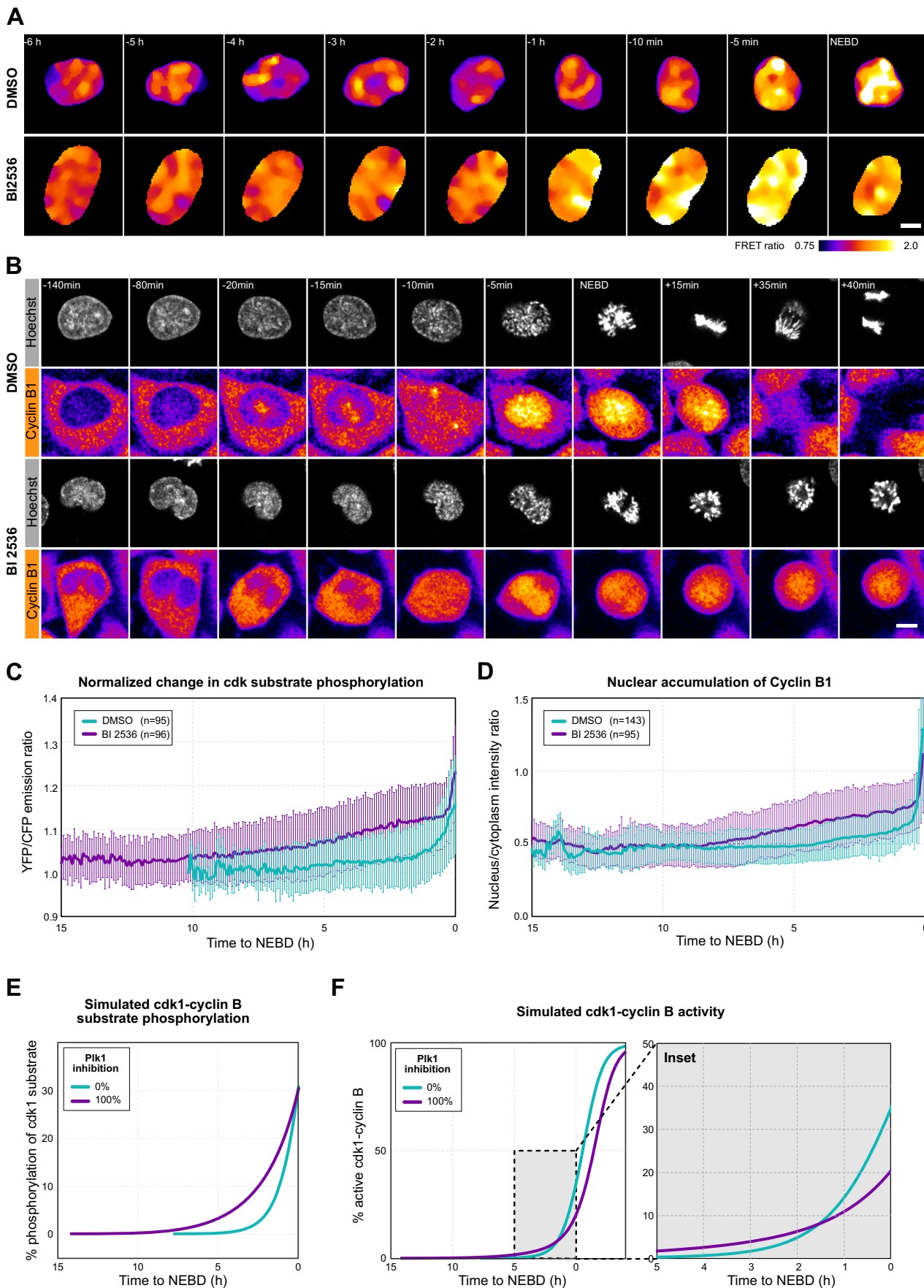

◄ **Figure 2. Cdk1 activity increases gradually over hours before NEBD.**

(A) Ratio images (YFP/CFP emission at CFP excitation) in a stable HeLa cell line expressing the Eevee-spCDK FRET probe. Selected frames showing cells entering mitosis. Time is relative to NEBD. (B) Live images of exemplary control and BI 2536 treated HeLa cells stably expressing Cyclin B1-mVenus while entering mitosis. Time is relative to NEBD. (C) Quantification of FRET ratio changes of the Eevee-spCDK FRET probe in control ($n = 95$) and BI 2536-treated ($n = 96$) populations of cells. Mean ± Standard Deviation (SD) is shown. (D) Quantification of Cyclin B1 nucleus to cytoplasm intensity ratio in control ($n = 143$) and BI 2536-treated ($n = 95$) populations of cells. Mean ± SD is shown. (E) Simulated fraction of cdk1 substrate phosphorylation ([$Subp$]) with and without plk1 inhibition, the activity of cdk2-cyclin A is [$CycACdk2_{Tot}$] = 4 a.u. (F) Simulated fraction of active cdk1-cyclin B, [$CycBCdk1$]/[$CycBCdk1_{Tot}$], with and without plk1 inhibition. For (A) and (B) scale bars are 10 μm. Source data are available online for this figure.

abolished by the cdk1 inhibitor RO-3306, confirming the specificity of the probe (Fig. EV2C). In BI 2536-treated cells we observed a much slower, continuous increase in the FRET signal indicative of a slow, gradual increase in cdk substrate phosphorylation over several hours (Figs. 2A,C and EV2A). Interestingly, cdk substrate phosphorylation reproducibly reached substantially higher levels compared to controls before cells transitioned to NEBD.

Cdk1 activation is also coupled to nuclear accumulation of cdk1-cyclin B1 during prophase. This has been shown to be critical for complete activation of cdk1, and to be required for driving the irreversible transition to prometaphase (Santos et al, 2012; Gavet and Pines, 2010). Therefore, we next visualized changes in the localization of cdk1-cyclin B1 by imaging HeLa cells expressing endogenously tagged cyclin B1-mVenus. In control cells, we could confirm the previously observed nuclear re-localization of cyclin B1 (Gavet and Pines, 2010), the kinetics of which closely followed activation of cdk1 starting ~30 min before NEBD (Fig. 2B,D, for single cell traces see EV2B). In BI 2536-treated cells, the slow and gradual increase in the level of cdk substrate phosphorylation detected by the FRET probe was mirrored by slow and gradual relocalization of cyclin B1 to the nucleus. We quantified these kinetics on the individual as well as on the cell population level, revealing a slow increase occurring over several hours and eventually reaching similar levels to control prometaphase cells at NEBD (Figs. 2B,D and EV2B).

To further confirm these results, we also stained the endogenous cyclin B1 protein by immunofluorescence in an untagged HeLa cell line and used chromosome morphology to stage cells. In the control sample we found cyclin B1 to localize to the cytoplasm in cells with uncondensed chromosomes (G2), while in cells with condensed chromosomes (prophase), cyclin B1 was enriched in the nucleus (Fig. EV2D,E). In the BI 2536-treated sample, we observed cytoplasmic localization in cells with non-condensed chromosomes, similar to controls. Different to controls, in cells with condensed chromosomes, we observed an equilibration of cyclin B1 between the cytoplasm and nucleus, with preferential nuclear localization in cells arrested in prolonged prophase for longer periods (17 h) (Fig. EV2D,E).

We next wanted to see if the mathematical model above can capture the observed gradual increase in cdk1 substrate phosphorylation detected by the FRET probe. Therefore, we simulated the phosphorylation state of the cdk1-cyclin B substrate, a proxy for the FRET-signal in the model (Fig. 2E). The simulations closely matched the kinetics observed experimentally by the cdk FRET probe in untreated cells and in plk1-inhibited cells (compare Fig. 2C,E). The simulations also capture the experimental observation that by the end of the long delay the amount of phosphorylated cdk1 substrate is substantially higher in plk1-inhibited cells than in controls.

Simulations show that this is due to a higher level of cdk1-cyclin B activity reached in plk1-inhibited cells before NEBD (Fig. 2F). The cause for this is that in the absence of plk1, the level of active cdc25 increases slower, and therefore a longer time and eventually higher cdk1 activity is required to achieve a sustained cdc25-cdk1 positive feedback. This feedback then drives the switch-like activation of cdk1.

Taken together, the cdk activity FRET probe and cyclin B1 localization jointly evidence that in plk1-inhibited cells cdk1 activity increases slowly and gradually over several hours preceding its full activation at NEBD. This behavior is recapitulated by simulations of the signaling network, together evidencing a unique, prolonged state early in mitosis with low and gradually increasing cdk1 activity in plk1-inhibited cells.

## Plk1 inhibited cells are delayed in a prolonged prophase-like state with condensing chromosomes

We next wanted to characterize this prolonged, low cdk1 activity state at the cellular level. In control cells, closer inspection of live recordings of Hoechst-stained cells revealed early signs of chromosome condensation, a hallmark of prophase, 10–20 min prior to NEBD. In BI 2536-treated cells we observed condensing chromosomes much earlier, several hours preceding NEBD in all three cell lines investigated (Fig. 3A). In order to quantify this effect, we automatically segmented nuclei and used the standard deviation of fluorescence intensities as a measure of chromosome condensation (Neurohr and Gerlich, 2009). By this measure, we were able to detect the start of chromosome condensation in control cells already much earlier, 1–2 h before NEBD: a slow and gradual increase followed by a steep rise ~20 min before NEBD (Fig. 3B). In BI 2536-treated cells we observed chromosome condensation to begin up to 10–15 h before NEBD. Chromosome condensation then continued up to NEBD, while several cells remained arrested with condensing chromosomes until the end of the recording (Fig. 3B). The timing differed slightly between cell lines (see above), but overall they all showed a similar response (Fig. 3A,B). HeLa cells treated with GSK 461364 also showed an identical phenotype displaying delayed mitotic entry with progressively condensing chromosomes (Fig. EV3A,C).

The above effects observed in single cells were also reflected on population averages quantified on all tracked cells in a field of view. Control cells showed no systematic trend in chromosome condensation during G2. By contrast, in BI 2536-treated cells, chromosome condensation progressively increased resulting from more and more cells starting to condense chromosomes in the population hours before NEBD (Fig. 3C). The increase in chromosome condensation observed in populations of live cells correlated tightly with a progressive increase in the number of cells

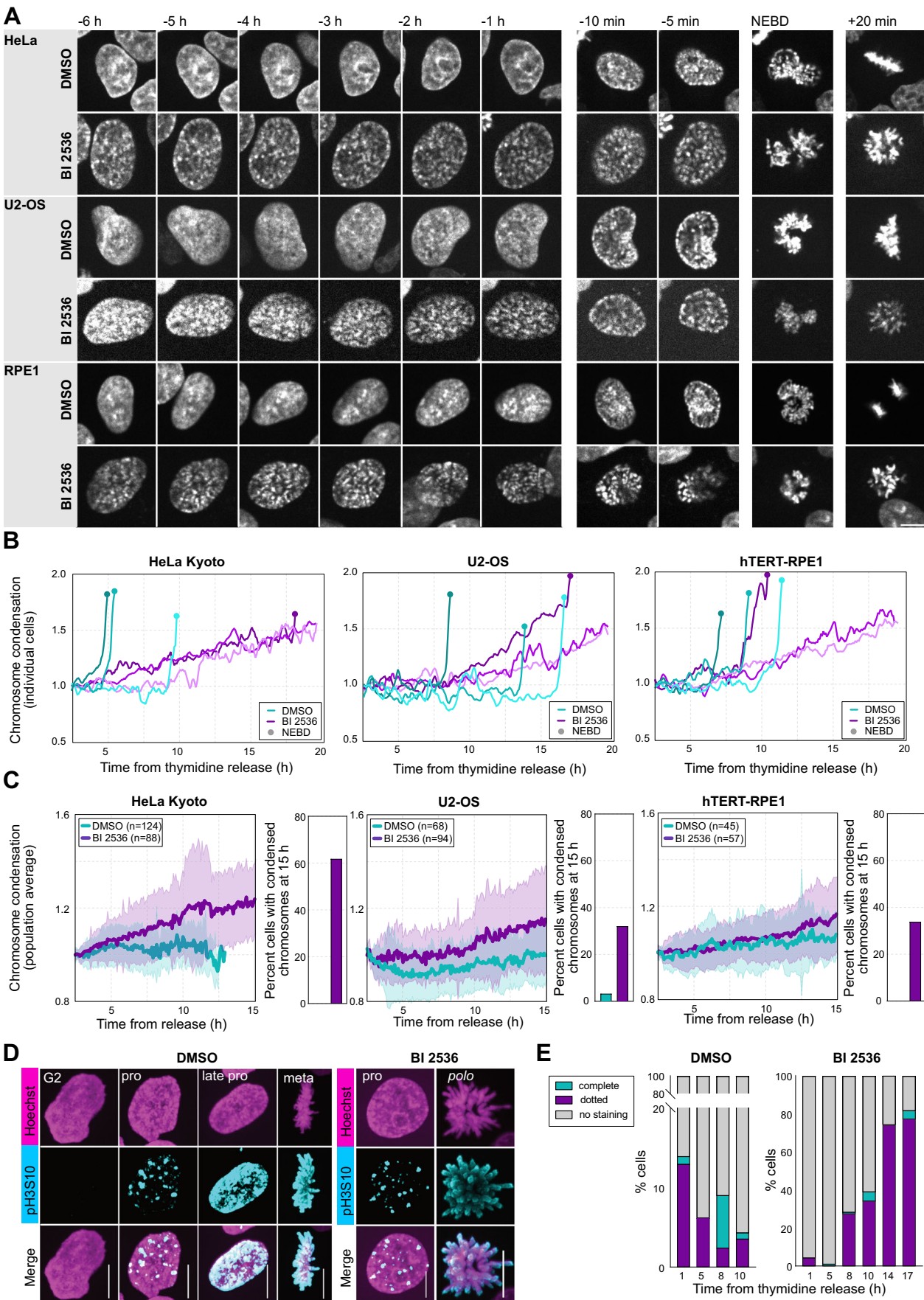

**Figure 3. Cells are delayed in a prolonged prophase-like state with condensing chromosomes.**

(A) Montages showing mitotic entry of exemplary control and BI 2536-treated cells (250 nM for HeLa and 100 nM for U2-OS and RPE1) stained with 5-SiR-Hoechst. Time is relative to NEBD. (B) Quantification of chromosome condensation using the standard deviation of fluorescence intensity of individual cells on recording similar to images shown in (A). (C) Population averages of tracked nuclei of measurements as in (B). Mean ± Standard Deviation (SD) is shown. (D) Immunofluorescence staining of pH3S10 in HeLa Kyoto cells treated with and without 250 nM BI 2536 at different stages of division. (E) Quantification of different morphologies on images similar to (D) in cells fixed at different times after thymidine release. The quantification is done only on attached cells. For (A) and (D) scale bars are 10 µm. Source data are available online for this figure.

showing phospho-histone 3 serine 10 (pH3S10) staining (Fig. 3D,E). At the late time point, 17 h after thymidine release, up to 80% of the BI 2536-treated cells showed the dotted pH3S10 staining pattern typical of prophase (Fig. 3E) (Hirota et al, 2005).

Taken together, our data show consistently across cell lines and structurally unrelated inhibitors that when plk1 is inhibited, cells are delayed in a prolonged prophase-like state with progressively condensing chromosomes displaying the early, dotted pH3S10 marks.

## Prolonged prophase cells display dynamic microtubules, signs of rounding up and intact nuclear envelope

Next we wanted to see whether delayed cells show hallmarks of prophase other than chromosome condensation. Due to their high synchronicity, suitability for live-imaging and easy handling, we performed subsequent experiments with HeLa (Kyoto) cells. We first visualized microtubules by using low concentrations of the fluorescently labeled taxane, 6-SiR-CTX (Bucevicius et al, 2020). Confirming previous reports, while BI 2536-treatment did not visibly change the total amount of microtubules in prophase and prometaphase cells, microtubule organization was strongly affected (Fig. 4A). As plk1 activity is required to recruit the pericentriolar material to centrosomes, in BI 2536-treated cells, centrosomal asters do not form, and the microtubule network shows a de-focused organization (Lénárt et al, 2007).

To assay the dynamics of these microtubules, we imaged cells stably expressing EB3-mEGFP at different phases of mitosis staged by chromosome morphology (Fig. 4B). We tracked microtubule tips using the plusTipTracker software to quantify microtubule dynamics (Applegate et al, 2011). In control cells, this analysis revealed dynamics consistent with data published earlier (Ferenz and Wadsworth, 2007), showing an increase in plus-tip velocities as cells progress from G2 to prophase and then to prometaphase (Fig. 4B). Cells delayed in prolonged prophase by BI 2536 showed plus-tip growth velocities significantly higher than prophase, and even higher than prometaphase of DMSO-treated control cells (Fig. 4B). Thus, cells in prolonged prophase not only display condensed chromosomes, but also show increased, mitotic-like microtubule dynamics.

Secondly, we monitored the dynamics of the actin cytoskeleton by imaging cells stably expressing the actin marker, LifeAct-mEGFP (Riedl et al, 2008). This marker displayed a strong cortical staining in BI 2536-treated cells delayed in prolonged prophase, comparable or even stronger than the cortical label in control cells at metaphase (Fig. 4C). Concomitantly, we observed attempts of BI 2536-treated cells to detach and to begin to round up 1–2 h before NEBD, much earlier than this occurs in control cells (Fig. 4C). We could see cells partially rounding up, and in several instances we

observed an oscillatory behavior with cells rounding up and then attaching again during prolonged prophase (arrows in Fig. 4D). To quantify these effects, we measured the eccentricity of the cell outlines (marked by LifeAct-EGFP). This showed a slow, continuous decrease in BI 2536-treated cells as opposed to the sudden decrease in eccentricity shortly before NEBD in control cells (Fig. 4E). The change in cell shape was also mirrored by a change in nuclear shape visible on z-sections and reflected by a continuous increase of the position of the centroid of the nucleus along the z-axis (Fig. 4F,G).

Next, we monitored the state of the nuclear envelope (NE) in cells delayed in prolonged prophase. To this end, we stably expressed the nuclear import substrate IBB-EGFP, the importin-β binding domain of importin-α fused to EGFP (Dultz et al, 2008). Confirming earlier reports, in DMSO-treated control cells, IBB-EGFP started to be released from the nucleus ~5 min before NEBD (Fig. 4H). BI 2536-treated cells showed no noticeable difference to controls by visual inspection, and as quantified by measuring the increase in IBB-EGFP intensity in the cytoplasm (Fig. 4I). Consistently, we did not observe a difference in the release kinetics of another NE marker, the inner nuclear membrane protein LBR (Fig. EV4A,B).

Together, these data indicate that mitotic reorganization of the actin cortex as well as the microtubule cytoskeleton begins during prolonged prophase, several hours before NEBD. Intriguingly, both microtubule dynamics and cortical actin appear to be activated beyond levels observed in control prophase, reaching as high as metaphase of control cells. This is consistent with our findings by the cdk FRET probe and computer simulations showing that cdk1 activity reaches higher levels in prolonged prophase in plk1-inhibited cells as compared to prophase in control cells. By contrast, the nuclear envelope remains intact and transport competent during prolonged prophase.

## Phosphoproteomic analysis of the prolonged prophase state

Combined, our data demonstrate that plk1 inhibition delays cells for several hours in a prophase-like state with low and slowly increasing cdk1 activity and an intact nucleus. This offers an opportunity, for the first time, to collect a large homogenous population of cells during early cell division and characterize this state by phosphoproteomics.

Therefore, we designed an experiment to quantitatively compare the following states (Fig. 5A): (i) attached cells in G2 5 h after release from a 24 h thymidine arrest; (ii) shaken-off Prometaphase cells arrested by the kinesin-5 inhibitor, S-trityl-L-cysteine (STLC) (Kaan et al, 2011) 13.5 h after thymidine release; and (iii) cells treated with 250 nM BI 2536 collected 13.5 h after thymidine

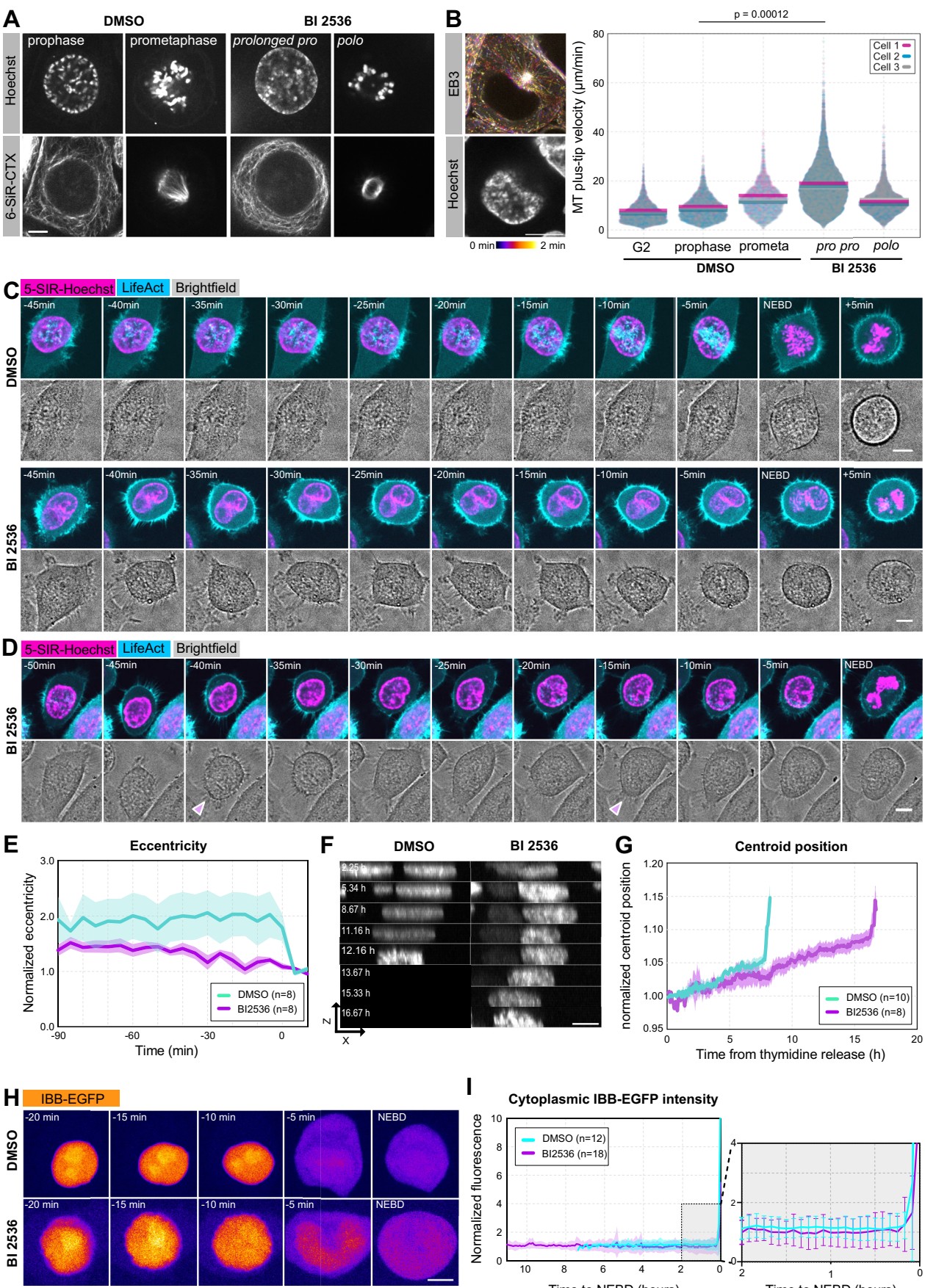

**Figure 4. Prolonged prophase cells display dynamic microtubules, signs of rounding up and an intact nuclear envelope.**

(A) HeLa cells stained with 5-580-CP Hoechst and 6-SiR-CTX at different stages of mitosis treated with or without 250 nM BI 2536. (B) Left: HeLa cells expressing EB3-mEGFP and stained with 5-SiR-Hoechst (EB3-mEGFP is shown as a time projection of 81 frames every 2 s). Right: Microtubule (MT) plus tip velocities of all EB3 tracks recorded from 3 cells at the indicated stages of division. Horizontal bars represent the mean for each cell. Unpaired Student's $t$ test was used for statistical significance assessment between the mean values of the track velocities belonging to prophase and *ProPro* cells ($n = 3$ for each) with $p = 0.00012$. (C) HeLa cells expressing LifeAct-mEGFP and stained with 5-SiR-Hoechst treated with and without BI 2536. Time is relative to NEBD. (D) Images of a BI 2536-treated HeLa cell before undergoing NEBD. Arrows indicate subsequent rounding-up events. (E) Measurement of the 2D-eccentricity of the LifeAct-mEGFP labeled cell outline in experiments similar to (C). Data are normalized to the mean eccentricity of the 2 frames after NEBD. Mean ± Standard Error of Mean (SEM) is shown. (F) XZ 'side view' of cells stained with 5-SiR-Hoechst. Time is relative to thymidine release and DMSO/BI 2536 treatment. (G) Quantification of data shown in (F) by calculation of the position of the centroid (center of mass) in the Z-axis normalized to the mean position in the first 5 frames. Mean ± Standard Error of Mean (SEM) is shown. (H) Selected frames from a time lapse showing the localization of IBB-EGFP in the time prior to NEBD in control and BI 2536-treated cells. Time is relative to NEBD. (I) Quantification of IBB-EGFP mean cytoplasmic fluorescence intensity in control and BI 536-treated cells on recordings similar to (H). Data are normalized to the mean intensity in the first 5 frames of imaging. Inset showing the time window around NEBD is shown on the right. Mean ± Standard Deviation (SD) is shown. For (A), (C), (D), (F), and (H): scale bars are 10 μm. Source data are available online for this figure.

release. This latter sample was further divided into two: after a gentle shake-off, suspended cells were collected separately from attached cells, resulting in *Polo* and Prolonged Prophase cell populations, respectively. As confirmed by Hoechst staining, the floating cells were almost exclusively (>98%) cells arrested in prometaphase showing the *Polo* phenotype (the others being mostly dead cells), while the vast majority (>85%) of attached cells were in Prolonged Prophase (the other 15% being mostly cells in G2). We prepared three biological replicates and used sixplex Tandem Mass Tag (TMTsixplex™) isobaric labeling to quantify relative differences in protein phosphorylation.

The phosphoproteomics experiment identified 14,994 phosphosites, of which, after filtering, selection, transformation and imputation of the dataset, 2164 phosphosites were kept for further analysis (Dataset EV1 and EV2). Phosphosite intensities were then normalized showing an overall similar distribution of intensities across biological replicates (Fig. EV5A). Based on peptides from the flow through fraction of phosphopeptide enrichment, we identified 7793 protein groups, of which 5479 were kept after filtering and imputation. We did not detect a significant change in the abundance of proteins harboring differentially phosphorylated sites.

Hierarchical clustering of the phosphoproteomes revealed a clear separation between pre-NEBD (G2 and Prolonged Prophase) and post-NEBD (Prometaphase and *Polo*) samples (Fig. 5B). Within these categories, replicates of each sample formed distinct clusters. This confirms, firstly, that the variability between replicates is substantially smaller than the difference between treatments. Secondly, the large difference between Prolonged Prophase and *Polo* samples additionally validates our approach to separate suspended *Polo* and attached Prolonged Prophase (*ProPro*) cells.

For comprehensive analysis of the four states, we employed linear modeling with a 2-by-2 factorial design using +/− BI 2536 treatment and mitotic time as the two axes (Fig. 5C). Along the time axis, we are comparing pre-NEBD to post-NEBD samples, i.e., G2 and Prolonged Prophase (*ProPro*) to the two mitotic, post-NEBD samples, STLC-arrested Prometaphase with and without BI 2536. Along the BI 2536 axis, we are comparing samples with and without plk1 inhibition.

We next explored how the four samples differ using four pairwise comparisons (Fig. 5C). Prometaphase (Prometa) and *Polo*

samples showed the typical signature of massive mitotic phosphorylation, reflected by the large number of changes detected when comparing each to its pre-NEBD counterpart (Fig. 5C). By contrast, pairwise comparison of the two mitotic samples, *Polo* and Prometa, revealed many known targets of plk1 (Fig. EV5B). Consistently, in this comparison, analysis of the differentially phosphorylated sites revealed plk1 as the main kinase predicted to phosphorylate these sites (Fig. EV5C). G2 and Prolonged Prophase (*ProPro*) samples lacked most mitotic phosphorylations, and only a few phosphosites were affected when directly comparing them against each other (Fig. 5C).

In order to identify phosphosites specific to Prolonged Prophase (*ProPro*), we established a fifth comparison (Fig. 5C). In this comparison, using the classic interaction model for a $2 \times 2$ factorial experimental design, we compared ProPro against Polo, then G2 against Prometa, and then we combined these two comparisons. Thereby, we identify the significant differences specific to Prolonged Prophase, i.e., phosphosites specific to plk1 inhibition *and* before NEBD. This excludes the phosphosites that are common to plk1 inhibition before and after NEBD. It also excludes the sites that are common to the two pre-NEBD samples (Fig. 5C). Below we focus our attention to these Prolonged-Prophase-specific phosphosites, but each of these categories contains a wealth of valuable information on the respective states that we report in Dataset EV3.

For the Prolonged-Prophase-specific hits, gene ontology (GO) analysis identified enriched terms related to nuclear organization, cytoskeleton and the cell cycle, an overall signature consistent with the phenotypic changes we observed in Prolonged Prophase (Fig. 5D). Of the 389 peptides differentially phosphorylated in Prolonged Prophase, 246 were phosphorylated and 143 were de-phosphorylated (Fig. 5E, Dataset EV3). Kinase enrichment analysis of hyperphosphorylated peptides in Prolonged Prophase identified primarily cyclin-dependent kinases, including cdk1 (Fig. EV5D).

Thus, by using BI 2536 to delay mitotic entry, we were able to harvest a highly enriched population of cells in Prolonged Prophase for phosphoproteomic analysis. By using linear modeling for quantitative analysis, we were able to derive the phosphosites that are specifically phosphorylated or dephosphorylated in Prolonged Prophase. This analysis revealed that Prolonged Prophase is a state very distinct from all other three states, featuring a limited and specific set of phosphosites.

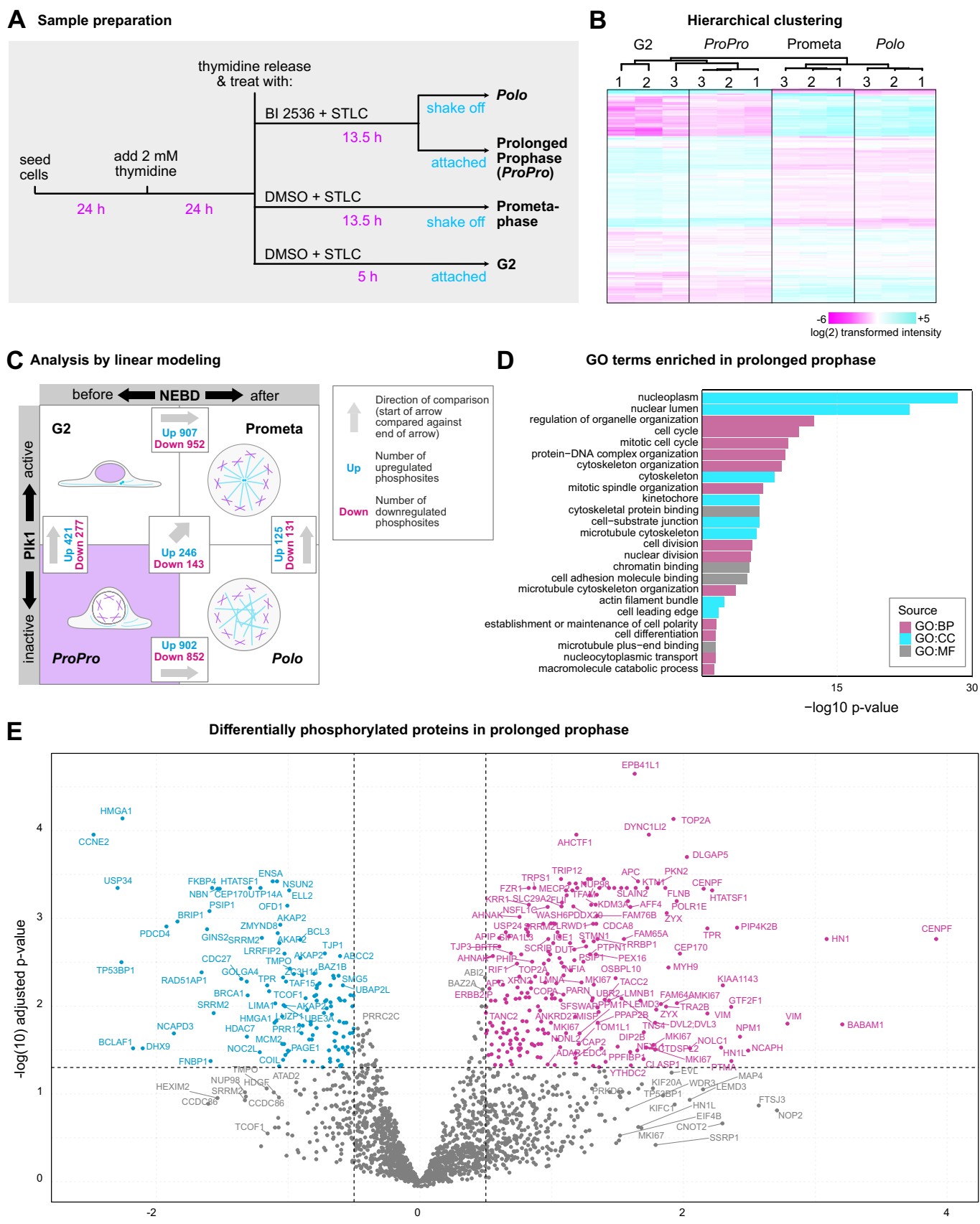

**Figure 5.  Phosphoproteomic profiling of prolonged prophase.**

(A) Experimental design of the proteomics experiment. Each sample had 3 biological replicates. (B) Hierarchical clustering of the normalized log(2) transformed intensities of the phosphosites from 4 samples and 3 replicates (denoted by 1, 2, 3). (C) Scheme showing the 2-by-2 factorial design upon which linear modeling was used for the analysis of the dataset. The number of phosphosites shown as Up refers to phosphosites detected by the indicated comparison with an adjusted *p*-value < 0.05 and log(2) fold change >0. The number of phosphosites shown as Down refers to phosphosites detected by the indicated comparison with an adjusted *p*-value < 0.05 and log(2) fold change <0. The arrow in the center refers to the fifth comparison to analyze phosphosites specific to ProPro. (D) Gene ontology terms enriched in the significantly hyperphosphorylated prolonged prophase-specific peptides detected after applying linear modeling. GO:BP, GO:CC, and GO:MF refer to Gene Ontology terms within the domains of Biological Process, Cellular Component, and Molecular Function, respectively. Testing for over representation for GO terms was conducted using the Hypergeometric test with a *p*-value cutoff of 0.05. (E) Volcano plot showing the differentially phosphorylated prolonged prophase specific peptides after applying linear modeling. Statistical significance was assessed using a moderated *t*-test where empirical Bayes moderation was applied to stabilize variance estimates across genes. The log(2) fold change threshold is set at 0.5 below and above 0 and the significance threshold is set at 0.05 adjusted *p*-value.

## Proteins phosphorylated in Prolonged Prophase are consistent with cellular phenotypes

We next used the STRING database to query the 326 proteins displaying the 389 phosphosites significantly altered in Prolonged Prophase. After clustering, we obtained the network shown in Appendix Fig. S1. The largest cluster with 44 nodes had the STRING description "Regulation of chromosome segregation - Cell Cycle Checkpoints" based on the cluster's consensus protein annotations (Fig. 6A; Appendix Table S1).

One of the most prominent changes in cellular organization we observed during Prolonged Prophase is the condensation of chromosomes. Indeed, we detected changes in phosphorylation on two different condensin subunits, NCAPH and NCAPD3 (Fig. 6A,B). Interestingly, these two subunits belong to two different complexes, condensin I and II, which localize differently to cytoplasm and nucleus, respectively (Hirota et al, 2004). Additionally, we detected changes on two phosphosites of topoisomerase-2α, a DNA de-catenating enzyme with well-established role in chromosome condensation (Nielsen et al, 2020). We also found one phosphosite on CDCA2/Repo-man (Vagnarelli et al, 2006), and prominently, we identified five significantly differentially phosphorylated peptides in KI-67, a protein that coats the surface of chromosomes and is required for their individualization (Cuylen et al, 2016). Additionally, we detected phosphopeptides in the centromere/kinetochore components CDCA8/Borealin and INCENP, two components of the chromosome passenger complex (Jelluma et al, 2008; Xu et al, 2009), and three phosphorylations on CENP-F, which in prometaphase has roles in stabilizing kinetochore microtubule attachments (Auckland et al, 2020). We also identified phosphorylation of the TTK/MPS1 kinase, which regulates the assembly of the mitotic checkpoint complex on the kinetochore (Diril et al, 2016; Hayward et al, 2019). Furthermore, we observed phosphorylation of the SWI/SNF and ISWI chromatin remodeling complexes, and histone modifying enzymes, which have been proposed to be involved in chromosome condensation (Wilkins et al, 2014). Taken together, our analysis reveals a set of proteins that are differentially phosphorylated in prophase with established or predicted roles in chromatin organization, chromosome condensation and individualization, as well as in centromere/kinetochore assembly.

More generally affecting nuclear organization, we also identified several nuclear envelope components to be differentially phosphorylated in Prolonged Prophase (Fig. 6A,B). This includes the lamins, lamin A and lamin B. We also identified additional proteins of the inner nuclear membrane, including the lamina-associated polypeptide 2 (LAP2, TMPO), Man1/LEMD3 and the Lim-domain protein (LMO7), a binding partner of Emerin. We found differential phosphorylations on proteins of the nuclear pore complex (NPC), specifically on ELYS, TPR, and Nup98. These changes indicate that besides the lamina, the associated protein network at the inner nuclear envelope as well as the nuclear pores begin to remodel during prophase. However, these changes are specific and restricted to a few sites as compared to the massive phosphorylation of these proteins in prometaphase. This is well consistent with the limited changes observed in nuclear envelope organization and nuclear transport during prophase, while the overall structure of the nuclear envelope remains intact at this stage (Dultz et al, 2008; Lénárt and Ellenberg, 2006). A complete disassembly of the nuclear envelope, nuclear pores and complete depolymerization of the lamina occurs at NEBD, as cells transition to prometaphase and cdk1 is fully activated (Beaudouin et al, 2002). Samejima and coworkers also found LAP2 and Man1 to be released early from chromatin in prophase in their analysis of chromatin associated proteins (Samejima et al, 2022). Furthermore, our data are also consistent with earlier work showing that phosphorylation of Nup98 (along with Nup53) is an early step in NPC disassembly during prophase (Linder et al, 2017).

Besides nuclear organization, changes in cytoskeletal dynamics are most prominent during Prolonged Prophase. Consistently, we identified many regulators of the microtubule cytoskeleton to be differentially phosphorylated (Fig. 6A,C). We identified phosphorylations on the centrosomal protein CEP170 (Conduit et al, 2015), as well as the centrosome associated microtubule regulator TACC2 (Peset and Vernos, 2008). Intriguingly, on the spindle assembly factor TPX2 we detected a phosphorylation on S738. This site has been identified and functionally characterized earlier as a critical 'early riser' prophase phosphorylation event (Ly et al, 2017). We additionally identified other regulators of microtubule assembly including CLASP1 and MAP4 (Samora et al, 2011), as well as CKAP5 (Ali et al, 2023) and SLAIN2 (van der Vaart et al, 2011). We also found differential phosphorylations on the microtubule disassembly factors Katanin and Stathmin (Cassimeris, 2002), which may function to sever long interphase microtubules. In addition, we identified differentially abundant phosphopeptides in motor proteins including regulators of kinesins and the intermediate light chain 2 of cytoplasmic dynein (DYNC1LI2), known to be involved in centrosome separation (Raaijmakers et al, 2012).

We also identified several regulators of the actin cytoskeleton as differentially phosphorylated in the Prolonged Prophase samples (Fig. 6A,C). We observed differential phosphorylations in the focal adhesion components zyxin, supervillin, and paxillin, and in a

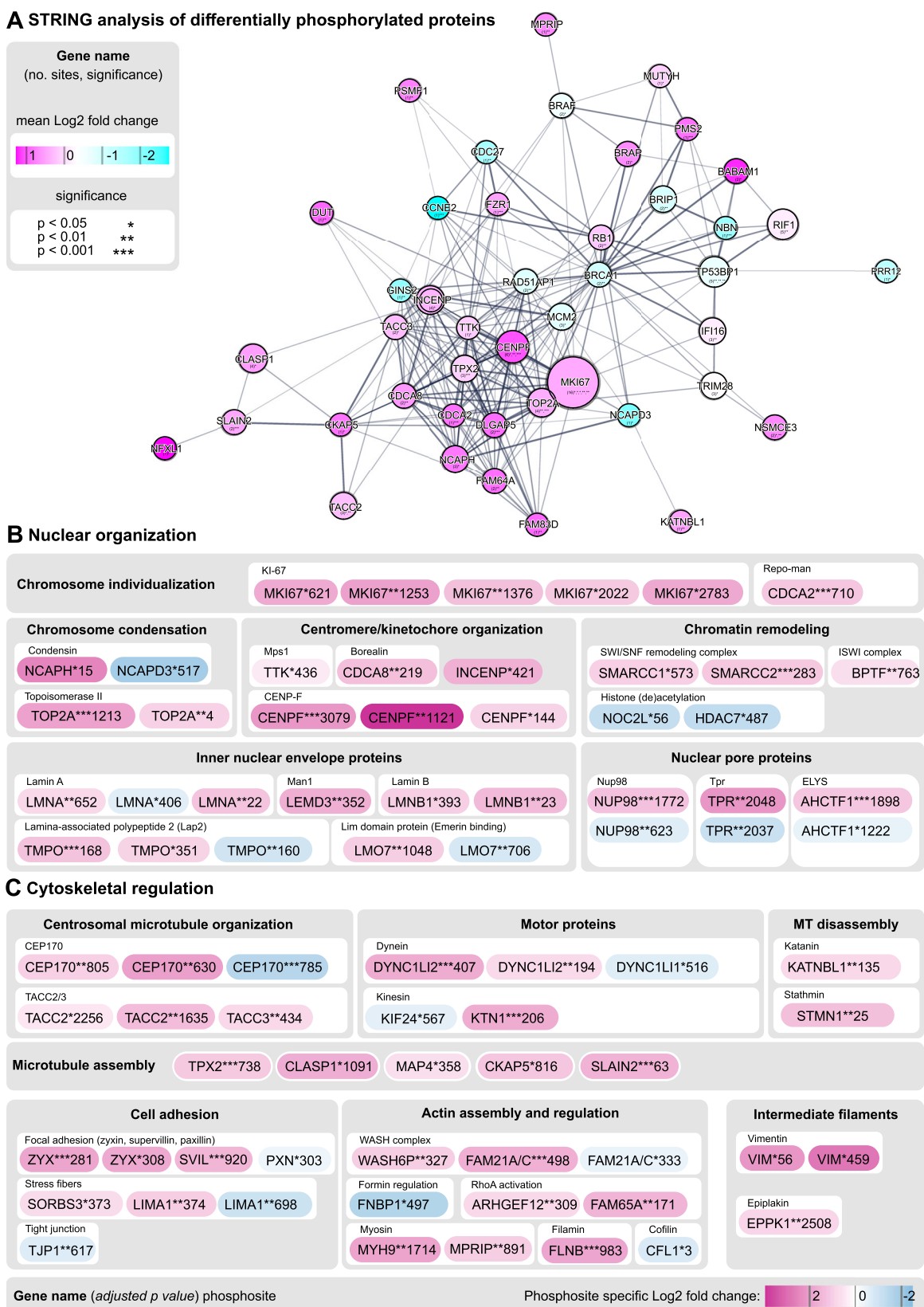

**A** STRING analysis of differentially phosphorylated proteins

**B** Nuclear organization

**C** Cytoskeletal regulation

**Figure 6.  Detailed analysis of the proteins phosphorylated in prolonged prophase.**

(A) Cluster network generated depicting differentially phosphorylated proteins specific to prolonged prophase after applying linear modeling. The gene names of the corresponding proteins are shown. The size of the nodes corresponds to the number of phosphosites detected per protein. The exact number of phosphosites detected is written inside of each node, with the asterisks marking the adjusted $p$-value of the significantly changed phosphosites (***, **, * corresponds to $p \leq 0.001$, 0.01, and 0.05, respectively). The color coding corresponds to the mean log(2) fold change of intensity values of all phosphosites. The line thickness indicates the strength of data support. (B, C) Differentially phosphorylated peptides specific to prolonged prophase after applying linear modeling grouped according to their cellular functions. The gene names of the corresponding peptides are shown, the asterisks mark the adjusted $p$-value with ***, **, * corresponding to $p \leq 0.001$, 0.01, and 0.05, respectively, followed by the position of the phosphosite within the amino acid sequence of the protein. The color coding corresponds to the log(2) fold change of intensity values. For (A–C), Statistical significance was assessed using a moderated $t$-test where empirical Bayes moderation was applied to stabilize variance estimates across genes.

regulator of stress fibers, LIMA1 (Chen et al, 2022). We also detected phosphosite-changes on components of tight junctions as well as intermediate filaments. Additionally, we detected changes indicative of activation of the Rho pathway, the main pathway responsible for increasing cortical contractility and tension as cells round up (Taubenberger et al, 2020). These proteins include the myosin phosphatase Rho interacting protein (MPRIP), the Rho guanine nucleotide exchange factor 12 (ARHGEF12) and the Rho family-interacting cell polarization regulator 1 (FAM65A/RIPOR1). Somewhat surprisingly, we also identified phosphorylation on several subunits of the WASH complex (FAM21A, FAM21C, WASH6P), an activator of the Arp2/3 actin nucleation complex (Rottner et al, 2010). WASH has been recently shown to be involved in nucleation of actin filaments at centrosomes (Farina et al, 2019), but whether this has a relevance in prophase remains unknown.

Together, we primarily detected phosphorylation of proteins involved in nuclear organization and on cytoskeletal regulators. This is well consistent with the observed condensation of chromosomes, the disassembly of interphase microtubules and actin structures (such as focal adhesions and stress fibers). In addition to the 44 proteins shown on Fig. 6, a complete list of differentially phosphorylated proteins are listed in Dataset EV3—a remarkably short list of proteins compared to the massive phosphorylation taking place in prometaphase. Thereby, this dataset provides a first insight into early mitotic phosphorylations that occur while cdk1 is only partially activated and together with other mitotic kinases mediate the first steps preparing the cell for division.

## Discussion

The role of plk1 in the regulation of mitotic entry has been studied earlier, but the specific stage at which plk1 inhibition delays cells during mitotic entry, and generally whether plk1 is essential for mitotic entry, remained somewhat contentious (Gheghiani et al, 2017; Lénárt et al, 2007). To find answers to these questions, we inhibited plk1 in three different cell lines using a wide range of concentrations of the highly-specific small-molecule inhibitors BI 2536 and GSK 461364. By using automated live-cell imaging, we could confirm that plk1 is critical for timely mitotic entry in all cell lines investigated. However, the effect is highly variable between cell lines as well as between individual cells: it ranges from a slight delay of a couple of hours to a complete block of mitotic entry.

We then performed simulations using a previously published mathematical model of the signaling network that controls mitotic entry (Rata et al, 2018). We integrated plk1 into the model, but otherwise we did not change the connectivity of the network, and only made minimal adjustments to the model parameters that did not affect the overall, bistable behavior of the signaling network. Simulations readily recapitulated the dose-dependent delay as well as the highly variable entry times upon plk1 inhibition. The latter can be explained by the high sensitivity of the network to small changes of initial parameters; for instance, a small change in initial cdk1/2 levels can increase the entry delay from 4–6 h to well over 24 h. This resolves the controversy whether plk1 is essential for mitotic entry, as a long delay of 24 h (the length of a complete cell cycle in HeLa cells) is perceived in an experiment as a complete block. However, simulations and consistent experimental data show that plk1 inhibition substantially delays mitotic entry, but it does not halt or arrest cells in a new steady state. While such a delay may appear phenotypically identical to a checkpoint-mediated arrest, mechanistically such a 'stop sign' is very distinct from the slowed progression caused by plk1 inhibition.

When we investigated the cell cycle state of these delayed cells, a FRET-based cdk sensor and nuclear relocalization of cyclin B consistently evidenced that cdk1 substrate phosphorylation gradually increases during the entry delay. This gradual increase occurs over several hours and reaches levels higher than in unperturbed prophase. The mathematical model recapitulates the slow gradual increase in cdk1 substrate phosphorylation levels, and shows that this results from a gradual increase in cdk1-cyclin B activity during the delay. The model also captures our experimental observations that by the end of the delay cdk1 activity is substantially higher than in unperturbed prophase. In the model, this results from the slowed accumulation of active cdc25 in plk1-inhibited cells that require higher cdk1 activity for sustaining the cdc25-cdk1 positive feedback. A large body of work primarily by the Novák group, and by others identified and experimentally validated the core signaling modules and the overall architecture of the signaling network (e.g., Hutter et al, 2017). However, it is important to bear in mind that these models contain simplifications and do not implement every single reaction. For example, the model does not explicitly consider the spatial changes, such as relocalization of components within the cell. It is well established that in prophase, early changes to the nuclear envelope facilitate nuclear accumulation of cdk1-cyclin B (Gavet and Pines, 2010), which is then key to full activation of cdk1, driving NEBD and transition to prometaphase (Santos et al, 2012). It is also known that plk1 phosphorylates the nucleoporins Nup98 and Nup53 facilitating the permeabilization of the nuclear envelope in prophase (Linder et al, 2017). Combined with the fact that, as we show, the nuclear envelope remains intact and transport competent in plk1-inhibited cells, it is therefore well possible that

plk1 inhibition additionally slows mitotic entry by affecting nuclear accumulation of cdk1-cyclin B. Thus, while this would leave the conclusions we drew from simulations unchanged, in reality plk1 may act through parallel and more complex paths to facilitate mitotic entry.

By analyzing cellular phenotypes, we could show that plk1 inhibition delays cells in a prophase-like state: cells show progressively condensing chromosomes, increased microtubule dynamics and remodeling of the actin cytoskeleton in preparation for rounding up. We observed that each of these phenotypes is more pronounced in plk1-inhibited cells: chromosomes are more condensed than in unperturbed prophase, microtubule dynamics reaches levels similar to metaphase and the actin cortex thickens to a similar extent to unperturbed metaphase. These observations were puzzling at first, but are fully consistent with the increased level of cdk1 substrate phosphorylation measured by the FRET sensor. Together, these experimental observations confirm predictions of the mathematical model and show that during the extended delay cdk1 substrates are phosphorylated to a higher degree than in unperturbed prophase.

The extended time period plk1-inhibited cells spend in the prolonged prophase state enabled us to analyze this state by phosphoproteomics. We carefully designed an experiment in which we were able to quantitatively compare this prolonged prophase to other stages preparing a set of samples before and after mitotic entry and with and without plk1 inhibition. By combined analysis of these samples using linear modeling, we derived the phospho-sites specific to the prolonged prophase state (note that we excluded plk1-specific sites present both before and after NEBD, i.e in prophase and prometaphase). This defined a relatively small set of phosphorylations, the majority of which are predicted to be cdk1 sites, distinct from the much larger number of phosphosites that are detected in prometaphase when cdk1 is fully activated. By both automated and manual functional annotation, the majority of these proteins are involved in nuclear organization and cytoskeletal regulation, well matching the observed changes in cellular architecture.

The above conclusions combined clarify the role of plk1 in mitotic entry. Plk1 inhibition slows the accumulation of active cdc25C that in turn slows the onset of the positive feedback responsible for full activation of cdk1. The mathematical model readily explains the observed high variability in the length of the delay, and resolves controversies that suspected technical/experimental problems as a cause. Our simulations combined with experiments also define for the first time the prolonged prophase-like state, in which plk1-inhibited cells spend several hours. In this state cdk1 is partially active and nucleo-cytoplasmic compartmentalization is still maintained, two key defining aspects of prophase. However, this state also differs from unperturbed prophase in that plk1 is inhibited, preventing centrosome maturation and phosphorylation of certain nuclear pore proteins, for example (Lee and Rhee, 2011; Linder et al, 2017). Additionally, we show that cdk1 activity reaches substantially higher levels in prolonged than in unperturbed prophase. It is important to keep these differences in mind, however, as we show, prolonged prophase cells do display early reorganization of the cellular architecture very similar to normal prophase. By phosphoproteomics we identify specific changes in phosphorylations fully consistent with these cellular phenotypes. Prolonged prophase thus serves as a model for an otherwise inaccessible transient stage of early mitosis, before cdk1 is fully activated and the nuclear envelope disassembles.

In conclusion, despite advances in the field, we are still far from a complete understanding of the sequence of events leading up to mitotic entry, one of the most critical transitions in a cell's lifetime. We need to understand how the duration of individual events is determined, how they depend on one another, and ultimately we need to identify all the molecular modifications that underlie these changes. Identification and characterization of the prolonged prophase state will greatly facilitate addressing these important questions, as it allows collection of a homogenous population of cells in a prophase-like state, to manipulate their mechanical environment, and profile them using cellular, molecular, and biophysical assays.

# Methods

### Reagents and tools table

| Reagent/Resource | Reference or Source | Identifier or Catalog Number |
|---|---|---|
| **Experimental models** | | |
| HeLa Kyoto-WT | Jan Ellenberg (EMBL, Heidelberg, Germany) | N/A |
| U2-OS | Jan Ellenberg (EMBL, Heidelberg, Germany) | N/A |
| HeLa Kyoto EB3-mEGFP-mCherry-CenpA | Daniel Gerlich (IMBA, Vienna, Austria) | N/A |
| HeLa Kyoto H2B-mCherry-mEGFP-PCNA | Daniel Gerlich (IMBA, Vienna, Austria) | N/A |
| HeLa Kyoto LifeAct-mEGFP | Timo Betz (University of Göttingen, Göttingen, Germany) | N/A |
| hTERT RPE-1 | Luis Pardo (MPI-NAT, Göttingen, Germany) | N/A |
| HeLa Kyoto Cyclin B1-mVenus | Jonathon Pines (ICR, London, England) | N/A |
| HeLa Kyoto IBB-EGFP | This study | N/A |
| HeLa Kyoto Eevee-spCDK-FRET | This study | N/A |
| **Recombinant DNA** | | |
| pEevee-spCDK-FRET | (Sugiyama et al, 2024) | N/A |
| pEGFP-N1-IBB-EGFP | Jan Ellenberg (EMBL, Heidelberg, Germany) | 1072 |
| **Antibodies** | | |
| rabbit anti-H3S10ph (IF) | Sigma-Aldrich, Merck | 06-570 |
| mouse anti-cyclin B1 (IF) | BD Biosciences | 554177 |
| mouse anti-phospho-Histone H2A.X (Ser139) (IF) | Sigma Aldrich, Merck | 05-636 |
| Alexa Fluor 594 Alpaca Anti-Rabbit IgG (IF) | Jackson ImmunoResearch | 611-585-215 |
| Alexa Fluor 594 Alpaca Anti-Mouse IgG (IF) | Jackson ImmunoResearch | 615-585-214 |

| Reagent/Resource | Reference or Source | Identifier or Catalog Number |
|---|---|---|
| Alexa Fluor 488 Alpaca Anti-Mouse IgG (IF) | Jackson ImmunoResearch | 615-545-214 |
| Rabbit anti-Cdc25C (WB) | Cell Signaling Technology | 4688 |
| Rabbit anti-Phospho-Cdc25C (Thr48) (WB) | Cell Signaling Technology | 9527 |
| Rabbit anti-Phospho-cdc2/cdk1 (Tyr15) (WB) | Cell Signaling Technology | 9111 |
| Mouse anti-CDK1 (WB) | Thermo Scientific | MA5-11472 |
| CF™770 Goat Anti-Rabbit IgG (H + L) (WB) | Biotium | 20484 |
| CF™680 Goat Anti-Mouse IgG (H + L) (WB) | Biotium | 20065 |
| **Chemicals, Enzymes and other reagents** | | |
| Halt™ Protease and Phosphatase Inhibitor Cocktail (100X) | Thermo Scientific | 78440 |
| BI 2536 | MedChemExpress | HY-50698 |
| GSK461364 | MedChemExpress | HY-50877 |
| RO 3306 | Sigma-Aldrich | SML0569 |
| 5-SiR-Hoechst | Gražvydas Lukinavičius, (MPI-NAT, Göttingen, Germany) | N/A |
| Hoechst 33342 | Santa Cruz Biotechnology | sc-391054 |
| 5-580CP-Hoechst | Gražvydas Lukinavičius, (MPI-NAT, Göttingen, Germany) | N/A |
| Thymidine | Sigma-Aldrich | T1895 |
| S-Trityl-L-Cystein (STLC) | Fisher Scientific | 15510654 |
| Dulbecco's Modified Eagle Medium (DMEM) | Gibco, Thermo Fisher Scientific | 31053028 |
| Fetal Bovine Serum | Gibco, Thermo Fisher Scientific | 10270106 |
| Sodium Pyruvate | Gibco, Thermo Fisher Scientific | 11360039 |
| L-Glutamine | Gibco, Thermo Fisher Scientific | A2916801 |
| DMEM/F-12, GlutaMAX™ | Gibco, Thermo Fisher Scientific | 10565018 |
| hygromycin B | Roth | CP12.1 |
| FluoroBrite DMEM | Gibco, Thermo Fisher Scientific | A18967-01 |
| 96-well plates with cover glass bottom | Zell Kontakt | 5242-20 |
| #1.5 18 mm round coverslips | Paul Marienfield EN | 0117580 |
| paraformaldehyde solution | Electron Microscopy Sciences | 15714 |

| Reagent/Resource | Reference or Source | Identifier or Catalog Number |
|---|---|---|
| BSA | Roth | 8076.2 |
| ProLong Diamond mounting medium | Invitrogen | 15468070 |
| Epredia™ SuperFrost™ glass slides | Fisher Scientific | AG00008032E01MNZ20 |
| Lipofectamine 3000 transfection kit | Invitrogen | L3000-008 |
| Pierce™ BCA Protein Assay Kit - Reducing Agent Compatible | Thermo Scientific | 23250 |
| Pierce™ Universal Nuclease | Thermo Scientific | 88700 |
| TMTsixplex™ | Thermo Scientific | 90066 |
| NuPage 4–12% Bis-Tris gradient gel | Thermo Scientific | NW04120BOX |
| NuPAGE™ MES SDS-running buffer | Thermo Scientific | NP0002 |
| PageRuler Plus Prestained Protein | Thermo Scientific | 26619 |
| Nitrocellulose Membranes, 0.45 µm | Thermo Scientific | 88025 |
| QIAshredder | QIAGEN | 79656 |
| Fisherbrand Cell Scrapers | Fisher Scientific | 11587692 |
| **Software** | | |
| Fiji/ImageJ | (Schindelin et al, 2012) | N/A |
| Knime | (Berthold et al, 2008) | N/A |
| Perseus | (Tyanova et al, 2016) | N/A |
| MaxQuant | (Cox and Mann, 2008) | N/A |
| Matlab | MathWorks Inc. | N/A |
| R Studio | http://www.rstudio.com/ | N/A |
| VisiView 5.0 | Visitron Systems | N/A |
| Excel | Microsoft | N/A |
| Affinity Publisher | Serif Europe | N/A |
| XppAut | https://sites.pitt.edu/~phase/bard/bardware/xpp/xpp.htm | Version 8.0 |
| **Other** | | |
| Visitron Systems spinning disk confocal microscope | Visitron Systens | N/A |
| OkoLab incubator | OkoLab | N/A |
| Bioruptor | Diagenode | N/A |
| Odyssey® DLx fluorescent imager | LI-COR | N/A |

## Mammalian cell culture and drug treatments

HeLa and U2-OS cells were cultured in Dulbecco's modified Eagle medium (DMEM) supplemented with 10% Fetal Bovine Serum (FBS), 1% of 100 mM Sodium Pyruvate, and 1% of 200 mM

L-Glutamine, all from Gibco, Thermo Fisher Scientific. hTERT RPE-1 cells were cultured in DMEM/F-12, GlutaMAX™ Supplement (Gibco, Thermo Fisher Scientific) supplemented with 10% FBS and 10 µg/ml hygromycin B. All cell lines tested negative for mycoplasma contamination and were authenticated using STR profiling.

For synchronization, cells were treated with 2 mM (HeLa, U2-OS) or 5 mM (hTERT RPE-1) thymidine in DMEM. The treatment was conducted either after at least four hours from seeding, after ensuring the cells were properly attached, or after 24 h. 24 h after the thymidine treatment, the medium was removed and fresh medium containing either Dimethylsulfoxide (DMSO), BI 2536 (MedChemExpress, pre-dissolved in DMSO), GSK461364 (MedChemExpress, pre-dissolved in DMSO), or RO 3306 (Sigma-Aldrich, pre-dissolved in DMSO) was added.

## Live-cell imaging experiments and dyes

Cells were imaged in FluoroBrite DMEM (Gibco, Thermo Fisher Scientific) supplemented as above. Typically, 10,000 or 12,000 cells were seeded in 96-well plates with cover glass bottom (Zell Kontakt). Dyes, such as 100 nM 5-SiR-Hoechst or 5-580CP-Hoechst (kind gifts from the Lukinavičius group, MPI-NAT in Göttingen (Lukinavičius et al, 2015; Bucevičius et al, 2019)) to label DNA, was added to the medium prior to imaging.

Imaging was done on either a Visitron Systems spinning disk confocal microscope based on a Nikon Ti2 microscope body and a Hamamatsu CSU-W1 scan head equipped with an OkoLab incubator set to 37 °C with 5% CO$_2$ with an airflow of 0.3 l/min, or a Zeiss LSM 880 laser scanning confocal microscope equipped with a Pecon incubator box set to 37 °C with 5% CO$_2$. A Nikon Plan Apo VC, 20x Air objective (NA 0.75), Nikon Plan Apo, 40x Air objective (NA 0.95), Nikon Plan Apo TIRF AC, 100x Oil objective (NA 1.49) or Zeiss Plan Apo 40x Air (NA 0.95) objective lens were used for imaging. The 458, 488, 514, 561, and 640 nm excitation lasers were used dependent on the experimental setup, with a frequency of imaging every 5 or 10 min with 2 z planes 5 µm apart. Exposure times and laser powers were optimized in prior experiments to ensure that imaging does not interfere with cell survival and normal division timing. All live-imaging experiments were replicated at least 3 times.

## Immunofluorescence experiments and antibodies

For immunofluorescence experiments, cells were seeded in DMEM without phenol red with supplements as above. Either 60,000 or 90,000 cells were seeded on #1.5 18 mm round coverslips (Paul Marienfeld EN) in a 12-well plate. Fixation was conducted using 3% formaldehyde solution (Electron Microscopy Sciences) in PHEM buffer (1.8% PIPES, 4.2% HEPES, 0.38% EGTA, and 0.05% MgSO$_4$.7H$_2$O, adjusted to pH 7.0 with KOH) for 10 min. Cells were then permeabilized using 0.5% Triton X-100 in 1xPBS for 10 min and subsequently washed twice with 1xPBS. For storage overnight, the coverslips were stored in 1xPBS solution containing 0.02% Sodium Azide. The next day, autofluorescence was quenched using 100 mM ammonium-chloride and 100 mM glycine in 1xPBS for 20 min. The coverslips were then washed with 0.1% Triton X-100 in 1xPBS and blocked using 3% BSA (Roth) in 0.1% Triton X-100 in 1xPBS for 30 min. Primary antibodies were incubated for

1 h. Next, coverslips were washed again with 0.1% Triton X-100 in 1xPBS and incubated with the secondary antibodies together with Hoechst for DNA staining for another 1 h in the dark. Following two additional washes with 0.1% Triton X-100 in 1xPBS and then 1xPBS, coverslips were mounted on SuperFrost slides using ProLong Diamond mounting medium (Invitrogen) and sealed with nail polish. Immunofluorescence samples were imaged on the same spinning disk microscope described above using the 40x Air or 100x Oil objective. All immunofluorescence experiments were replicated at least 3 times.

## Image analysis

ND files (VisiView 5.0, Visitron Systems) were opened using the Bio-Formats plugin by Open Microscopy Environment (OME) in Fiji/ImageJ (Schindelin et al, 2012). The z planes were typically maximum intensity projected. To produce a label image, movies were downsampled by half, denoised with Gaussian blurring (sigma of 3), and used as input to the StarDist plugin (Schmidt Uweand Weigert, 2018; Weigert et al, 2020) with the fluorescent nuclei model. Later, label images were tracked using the Trackmate plugin in Fiji (Tinevez et al, 2017; Ershov et al, 2022).

For quantifying mitotic entry, the number of nuclei in the region were counted on the first frame, then with each NEBD event, the number of cells that have undergone NEBD was counted and the cumulative percentage was calculated.

For measuring the number of gH2ax foci, the Find Maxima function of ImageJ was used with a prominence >100, and all spots detected within each nuclei were counted.

For quantifying chromosome condensation, images and labels were used as input to Knime (Berthold et al, 2008). In the Knime workflow, labels smaller than 1000 pixels and segments touching the borders were removed. After renaming labels to match each nucleus, several features were calculated for each nucleus, including the standard deviation of fluorescence intensity and the mean fluorescence intensity. The global background was subtracted from the mean intensity of each nucleus at all time frames, and then the standard deviation was divided by the outcome to correct for bleaching.

For measuring IBB, LBR, and Cyclin B1 nuclear and cytoplasmic intensities, the same Knime workflow was used with modifications. For generating cytoplasmic labels, nuclear labels were dilated by 10 pixels using the Morphological Labeling Operations node, and then a cytoplasmic "ring" label was produced using the Voronoi Segmentation and by subtracting the nuclei label from the 10-pixel dilated-nuclei label. For IBB and LBR, the mean of the cytoplasm subtracted by the global background was normalized to the mean of the fluorescence intensity in the cytoplasm in the first 5 frames.

For measuring Eevee-spCDK biosensor FRET, the same Knime workflow was used with some modifications, where the YFP and CFP mean fluorescence intensities were measured, background subtracted, and divided by each other at every time frame for each nucleus.

For quantifying microtubule dynamics using EB3 comets, we used the uTrack software (Jaqaman et al, 2008) and its PlusTipTracker package (Applegate et al, 2011) in MatLab (Math-Works Inc., Version 9.11 (R2021b)). The 2D microtubule plus-ends object tracking was used for comet detection with a maximum of 4 frames gap-closing.

For quantifying centroid position, cells were imaged over time with 8 Z-steps and 1 µm step size. Single cells were cropped, and from the maximum Z-projection of their movies, we created label images and tracked the labels over time using the TrackMate plugin (Tinevez et al, 2017; Ershov et al, 2022). We then measured the centroid position, where the integrated density of the area inside the mask was measured in all Z-positions, multiplied by the number of Z-step, and the sum of all the products of all Z-steps was divided by the sum of the integrated densities from the Z-steps. The centroid positions over time were normalized to the average position in the first 5 frames of imaging.

For quantifying LifeAct-GFP signal, masks of LifeAct signal were created by Gaussian blurring (sigma 3) and later thresholding using the Huang method (Huang and Wang, 1995). The holes in the masks were filled and the watershed method was used to separate touching cells. At last, using the previously described workflow in Knime, the eccentricity was measured inside of these masks around the time of NEBD and normalized to the mean of these values in the 10 min following NEBD.

All mathematical operations applied on the data and data alignment and organization were conducted using Excel (Microsoft), plots and statistical tests were produced using GraphPad Prism (GraphPad Software) or the R statistical programming language. Figures were created using Affinity Publisher.

## Phosphoproteomics

Four different samples of HeLa cells as shown in Fig. 5 were collected, each in three biological replicates. Cells were pelleted and lysed in SDS lysis buffer (4% [w/v] SDS, 150 mM Hepes-NaOH pH 7.5, 1x Halt™ Protease and Phosphatase Inhibitor Cocktail (Thermo Scientific)) by incubation for 5 min at 99 °C followed by sonication (alternating 30 s on- and 30 s off-cycles at highest intensity output (Bioruptor, Diagenode)); and by another incubation for 5 min at 99 °C.

Protein concentrations were determined using Pierce™ BCA Protein Assay Kit - Reducing Agent Compatible (Thermo Scientific). Proteins were reduced with 10 mM dithiothreitol (DTT) for 30 min at 37 °C, 300 rpm, and alkylated with 40 mM iodoacetamide (IAA) for 30 min at 25 °C, 300 rpm, in the dark. The reaction was quenched with 10 mM DTT for 5 min at 25 °C, 300 rpm. Aliquots containing 0.4 mg of protein for each replicate of each sample were further processed. Samples were diluted to final 0.4% [w/v] SDS and 1 mM $MgCl_2$ was added. DNA content was digested using 1250U of Pierce™ Universal Nuclease (Thermo Scientific) for 2 h at 37 °C, 300 rpm. For further processing, aliquots of 240 µg per sample were taken and all samples were pooled in equal protein amounts to serve as reference samples in TMT multi-batch normalisations.

Separate and pooled samples were cleaned up by single-pot, solid-phase-enhanced sample preparation protocol by (Hughes et al, 2019). Briefly, carboxylate modified magnetic beads (Cytiva) were added at a 1:10 protein-to-bead mass ratio and an equal volume of 100% EtOH was added. Beads were washed three times with 80% [v/v] EtOH. Proteins were digested in 50 mM TEAB containing trypsin (Promega) at a 1:20 enzyme-to-protein ratio overnight at 37 °C, 1000 rpm. Peptides were recovered from the magnetic beads and beads were washed once with 100 µl water. Pooled samples were split into six individual reference samples and further processed separately. Samples were dried in a speed vac concentrator to minimal volume (~10 µl). TMTsixplex™ (Thermo

Scientific) labeling was performed using three labeling batches according to manufacturer's instructions. Briefly, 41 µl of 100% acetonitrile (ACN) were added to dried labels and dissolved by occasional vortexing. Labeling reagents were transferred to sample vials, following incubation at room temperature, 1 h; and quenched with 8 µl of 5% [v/v] hydroxylamine for 15 min at room temperature. Samples were combined including pooled reference channels. 1/20 of pooled samples were separately dried in a speed vac concentrator for whole proteome analysis.

Residual samples were subjected to phosphopeptide enrichment according to the EasyPhos protocol (Humphrey et al, 2018). Briefly, sample volumes were adjusted to 800 µl with water and subsequently to final 228 mM KCl, 3.9 mM $KH_2PO_4$, 38% [v/v] ACN, 4.5% [v/v] trifluoroacetic acid (TFA). 15 mg $TiO_2$ beads (GL Sciences) in 80% [v/v] ACN, 6% [v/v] TFA were added (beads-to-protein ratio of >10:1). Samples were incubated for 20 min at 40 °C, 1000 rpm; and washed in 60% [v/v] ACN, 1% [v/v] TFA four times and finally resuspended in 80% [v/v] ACN, 0.5% [v/v] acetic acid and mounted on top of empty columns (Harvard Apparatus). Phosphopeptides were eluted in two steps with 40% [v/v] ACN, 15% [v/v] $NH_4OH$. Samples were snap-frozen in liquid nitrogen.

Whole proteome and phosphopeptide samples were cleaned-up using C18 MicroSpin columns (Harvard Apparatus) according to manufacturer's instructions. Briefly, columns were equilibrated using 100% ACN; 50% [v/v] ACN, 0.1% [v/v] formic acid (FA); 0.1% [v/v] FA. Samples were loaded twice, washed three times with 0.1% [v/v] FA, and eluted twice with 50% [v/v] ACN, 0.1% [v/v] FA and once with 80% [v/v] ACN, 0.1% [v/v] FA. Eluants were dried in a speed vac concentrator and dissolved in 10 mM $NH_4OH$ pH 10, 5% [v/v] ACN for separation on an Xbridge C18 column (Waters) using an Agilent 1100 series chromatography system. The column was operated at a flow rate of 60 µl/min with a buffer system of buffer A (10 mM $NH_4OH$ pH 10) and buffer B (10 mM $NH_4OH$ pH 10, 80% [v/v] ACN). Peptide separation was performed over 64 min using the following gradient: 5% B (0–7 min), 8–30% B (8–42 min), 30–50% B (43–50 min), 90–95% B (51–56 min), 5% B (57–64 min). The first 6 min were collected as flow-through (FT), followed by 48 × 1 min fractions, which were reduced to 12 fractions by concatenated pooling. Fractionated whole proteome and phosphopeptide samples were dried in a speed vac concentrator and subjected to LC-MS/MS analysis.

## LC-MS/MS analysis

Dried peptide samples were dissolved in 2% [v/v] ACN, 0.1% [v/v] TFA and injected onto a C18 PepMap100-trapping column (Thermo Scientific, 0.3 × 5 mm, 5 µm) connected to an in-house packed C18 analytical column (Dr Maisch GmbH, 75 µm × 300 mm). Columns were equilibrated using 98% buffer A (0.1% [v/v] FA), 2% buffer B (80% [v/v] ACN, 0.1% [v/v] FA). Liquid chromatography was performed using an UltiMate-3000 RSLC nanosystem (Thermo Scientific). Phosphopeptides were analyzed for 60 min using a two-step linear gradient (5% to 34% buffer B in 34 min; 34% to 50% buffer B in 10 min) followed by a 5 min washing step at 90% of buffer B. Whole proteome peptides were analyzed for 120 min, and a buffer B linear gradient of 7% to 45% over 105 min was applied followed by washing at 90% buffer B for 5 min. Eluting peptides were analyzed using an Orbitrap Fusion Lumos Tribrid Mass Spectrometer (Thermo Scientific). A synchronous precursor selection (SPS) MS3 method was used with the

following MS settings for phosphopeptides: MS1 scan range, 350–2000 *m/z*; MS1 resolution, 120,000 FWHM; AGC target MS1, 5E5; maximum injection time MS1, 50 ms; collision energy 1, 35%; charge states, 2+ to 7+; dynamic exclusion, 20 s; MS2 resolution, 30,000 FWHM, AGC target MS2, 5e5; maximum injection time MS2, 90 ms; fixed first mass MS2, *m/z* 132; MS3 resolution, 60,000 FWHM; collision energy 2, 45%; number of notches, 10; AGC target MS3, 5e5; maximum injection time MS3, 118 ms; fixed first mass MS3, *m/z* 120. For the analysis of whole proteome peptides, the following MS settings were used: MS1 scan range, 350–2000 *m/z*; MS1 resolution, 120,000 FWHM; AGC target MS1, 5E5; maximum injection time MS1, 50 ms; collision energy 1, 38%; charge states, 2+ to 7+; dynamic exclusion, 40 s; MS2 resolution, 15,000 FWHM, AGC target MS2, 2.5e5; maximum injection time MS2, 40 ms; fixed first mass MS2, *m/z* 132; MS3 resolution, 60,000 FWHM; collision energy, 45%; number of notches, 10; AGC target MS3, 2.5e5; maximum injection time MS3, 118 ms; fixed first mass MS3, *m/z* 120.

## Phosphoproteomics data analysis

MS raw files were processed using MaxQuant version 2.0.3.0 (Cox and Mann, 2008) with default settings except for: fixed modification, carbamidomethylation (C); variable modifications, oxidation (M), acetylation (N-term), phosphorylation (S,T,Y) (only for phosphopeptide data). MS3 reporter ions (TMTsixplex™) were selected as quantification type, defining pooled sample channels as reference channels. Human canonical protein sequences (UP000005640) were downloaded from UniProt knowledgebase (Reviewed sequences, Swiss-Prot; date of download: 11.07.2022; number of protein sequences: 20,598).

Following steps in data analysis were conducted in Perseus (Tyanova et al, 2016). For each phosphorylation site, a leading protein was selected based on the list of potential candidate proteins for this site reported by MaxQuant. The official gene name and Uniprot accession of the leading protein were used in the subsequent analyses. Reporter ion intensities for each phosphorylation site identified at one of the three multiplicity levels were extracted from MaxQuant "Phospho(S, T, Y).txt" output table. Potential contaminants, reversed sequences, and phosphorylation sites identified with localization probability <0.75 as determined by MaxQuant were excluded from further analysis. If a phosphorylation event contained <70% nonzero (valid) intensity values, the event was considered as not quantified in this labeling experiment. Otherwise, missing values were imputed for each TMT channel/replicate individually by random sampling from a Gaussian distribution with a width of 0.3 and a downshift of 1.8 of the log2-transformed intensities. Log2-transformed reporter ion intensities were then normalized using Tukey median polishing individually for each labeling experiment in R statistical programming language and later subjected to statistical testing using the Limma package (Ritchie et al, 2015).

Linear models were fitted using least squares method from Limma R library. *P*-values were estimated with the empirical Bayes method and multiple hypothesis testing correction was done with the Benjamini and Hochberg method. A significance cutoff of 0.05 adjusted *p*-value was used.

To extract phosphosites specific to prolonged prophase, treatment-contrast parameterization was established using a 2-by-2 factorial design

for the linear modeling, with the following resulting comparison [BI2536.attached - BI2536.suspended] - [DMSO.attached - DMSO.suspended]. In this comparison, we used the Classic Interaction Model that is incorporated in the Limma package to find the interaction between two comparisons.

For the protein-protein interaction network, all proteins with significantly affected phosphosites in this comparison were used to conduct a search on the STRING database, version 12.0 (Szklarczyk et al, 2023) with a minimum interaction score of 0.400 (medium confidence). MCL clustering with an inflation parameter of 1.9 was further applied to the network (Appendix Fig. S1) and a cluster with 44 nodes "Regulation of chromosome segregation - Cell Cycle Checkpoints" was further explored. The cluster is shown (Fig. 6A) with color coding corresponding to the mean phosphorylation level (Log2 fold change) to all phosphosites belonging to the proteins shown and with the size of the nodes corresponding to the number of phosphosites detected, which is also written inside of the nodes, with the asterisk following corresponding to the significance level of the significantly changed phosphosites. The numerical information relating to the figure is shown in Appendix Table S1.

For the kinase enrichment analysis, all the phosphopeptides tested for differential expression, including insignificant hits, were converted to the asterisk format, where the phosphorylated amino acid is marked with an asterisk (*) after the phospho-acceptor (S, T, Y). Peptide sequences, log-fold changes and *p*-values of differential expression were submitted for kinase prediction at phosphosite.org (Hornbeck et al, 2015) KinaseLibraryAction functional enrichment analysis based on differential expression. A *p*-value threshold of 0.05 was used to discriminate between foreground and the background sets.

Other R packages were also used to create the graphs shown, including ggplot2 (Wickham, 2009), GO.db, GOstats, and BioConductor (Huber et al, 2015).

## Mathematical modeling and simulation

The reaction network shown in Fig. EV1E is based on the previously published model from Rata et al (2018). We did not change any of the reactions and the set of ordinary differential equations that describe these reactions, with the exception of incorporating plk1. Based on published experimental work (Gheghiani et al, 2017), we included a plk1 term in the equations for cdc25. Cdc25 is phosphorylated by cdk1-cyclin B and plk1 and dephosphorylated by PP2A-B55 and a constitutive phosphatase. Plk1 is activated by (indirect) phosphorylation downstream of cdk2-cyclin A. This leads to the following modified equations for the non-phosphorylated and bi-phosphorylated cdc25:

$$\frac{d[Cdc25]}{dt} = (k_{PPX,Y15} + k_{B55,Cdc25}[PP2AB55])\,[Cdc25p]$$

$$- (k_{Cdk1,Cdc25}\,V_{Cdk1} + k_{Plk1,Cdc25}\,[Plk1])\,[Cdc25]$$

$$\frac{d[Cdc25pp]}{dt} = -(k_{PPX,Y15} + k_{B55,Cdc25}[PP2AB55])\,[Cdc25pp]$$

$$+ (k_{Cdk1,Cdc25}\,V_{Cdk1} + k_{Plk1,Cdc25}\,[Plk1])\,[Cdc25p]$$

The monophosphorylated cdc25 is given by the conservation relation:

$$[Cdc25p] = [Cdc25_{Tot}] - [Cdc25] - [Cdc25pp]$$

Furthermore we define $V_{Cdk1} = [CycBCdk1]$, and $[Plk1] = f_{Plk1}[CycACdk2_{Tot}]$, where $[CycACdk2_{Tot}]$ is the concentration of active cdk2-cyclin A. The parameter $f_{Plk1}$ varies between 1 and 0 between fully active and completely inhibited plk1 by BI 2536. As in the published work, the onset of mitosis, i.e., NEBD, is assumed to happen when a substrate of cdk1-cyclin B:

$$\frac{d[Subp]}{dt} = k_{Cdk1,Sub} V_{Cdk1}([Sub_{Tot}] - [Subp]) - k_{B55,Sub} [PP2AB55] [Subp]$$

reaches 30% phosphorylation.

The time from end of S-phase to NEBD (Fig. EV1C) is comparable to the time from thymidine release to NEBD (Fig. 1C). We therefore neglect the remaining S-phase after thymidine release and assume that cells directly enter G2. G2 entry is modeled by a step increase in $[CycACdk2_{Tot}]$ from 0 to a positive value.

We first performed simulations using the originally published set of parameters. However, using the original parameter set NEBD occurred within 10–30 min after G2 entry, which is not a realistic timescale for mammalian cells, which typically require hours to progress through these stages. Therefore, we scaled parameter values to qualitatively match the timescale of our experimental system, i.e., lowered all rate constants by a constant factor, with the exception of the rate constants controlling cdc25 and cdk1-dependent substrate phosphorylation (Table EV1). For $[CycACdk2_{Tot}]$, we choose a value such that cells undergo NEBD when plk1 is fully inhibited ($f_{Plk1} = 0$), again matching experimental observations. The rate constants of cdc25 and cdk1-dependent substrate phosphorylation were set to recapitulate the early increase in cdk1 substrate phosphorylation we document here using the cdk FRET probe (Fig. 2A,C), and consistent with data published earlier (Gavet and Pines, 2010). In essence, we slightly slowed the positive feedback of cdk1 on cdc25 and thereby created a smoother activation of the feedback.

The simulations of NEBD entry (Figs. 1E and 2E,F) were performed using Matlab. The bifurcation diagram was computed using xpp-auto.

## SDS-PAGE and western blot

Samples of HeLa cells were collected as shown in Fig. EV1F. The shake-off samples were pelleted and lysed in SDS lysis buffer (4% [w/v] SDS, 150 mM Hepes-NaOH pH 7.5, 1x Halt™ Protease-Inhibitor-Cocktail (Thermo Scientific)). The attached samples were scrapped off the flask using Fisherbrand Cell Scrapers (Fisher Scientific) and then collected using the SDS lysis buffer. The lysates were then applied to QIAshredder columns (QIAGEN) followed by an incubation for 10 min at 95 °C and stored at −20 °C. Protein concentrations were determined using both Pierce™ BCA Protein Assay Kit - Reducing Agent Compatible (Thermo Scientific) and later using alpha-tubulin. Equal amounts of protein per sample were mixed with the sample loading buffer (10% SDS, 20% glycerol, 120 mM Tris pH 6.8, 10% beta-mercaptoethanol, and bromophenol blue) and were then incubated for 10 min at 95 °C. Next, the samples were loaded in a NuPage 4–12% Bis-Tris gradient gels (Thermo Scientific), with 1X NuPAGE™ MES SDS-running buffer

(Thermo Scientific). PageRuler Plus Prestained Protein was used as a protein ladder marker (Thermo Scientific).

Protein transfer onto nitrocellulose membranes, with pore diameters of 0.45 μm (Thermo Scientific) was conducted in a wet blotting chamber with a transfer solution of 25 mM Tris, 192 mM glycine, and 10% methanol for 60 min at 100 V. After transfer, the membranes were documented by staining with Ponceau S (Thermo Scientific). When needed, the blots were then destained with 0.1 M NaOH. The blots were then blocked for 1 h in TBST (Tris-buffered saline with 0.1% Tween-20) supplemented with 5% BSA (Bovine Serum Albumin Fraction V) (Roth). The blots were incubated overnight with primary antibodies and later for one hour with fluorescently-labeled secondary antibodies. The blots were then imaged using LI-COR Odyssey® DLx fluorescent imager. All western blot experiments were replicated at least 3 times.

## Data availability

The mass spectrometry data have been deposited to the ProteomeXchange Consortium (http://proteomecentral.proteomexchange.org) via the PRIDE partner (Perez-Riverol et al, 2025) repository with the dataset identifier PXD060406. The imaging dataset and image analysis numerical results, modeling computer scripts, and phosphoproteomics limma analysis code are deposited at: https://doi.org/10.17617/3.OYMBJ9.

The source data of this paper are collected in the following database record: biostudies:S-SCDT-10_1038-S44318-025-00400-9.

## Peer review information

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

## Acknowledgements

We would like to thank Jan Ellenberg (EMBL, Heidelberg, Germany), Daniel Gerlich (IMBA, Vienna, Austria), Luis Pardo (MPI-NAT), Jonathon Pines (Institute of Cancer Research, London, England) and Timo Betz (University of Göttingen) for providing cell lines, as well as Grazvydas Lukinavičius (MPI-NAT), Kazuhiro Aoki (NIBB, Okazaki, Japan), and Jochen Rink (MPI-NAT) for reagents. We would like to thank Alexander Stein for access to the western blot imaging system. We would like to thank Olexandr Dybkov and Ralf Pflanz for their help with the proteomics database. We would like to thank Pooja Mehta, Yehor Horokhovskyi, Juliane Liepe (MPI-NAT), Arshad Desai (Ludvig Cancer Research, UCSD, USA), Matthias Dobbelstein (Molecular Oncology, UMG, Göttingen, Germany) and Jonathon Pines (Institute of Cancer Research, London, UK) for comments on the manuscript. We would like to acknowledge the support of MPI-NAT's core facilities, the Synthetic Chemistry and the Proteomics core facilities in particular. HU was supported by a grant from the Deutsche Forschungsgemeinschaft (DFG) via SFB1565 (project number: 469281184). Research in PL's group is funded by the Max Planck Society, MG is enrolled in the IMPRS Molecular Biology graduate program.

## Author contributions

**Monica Gobran**: Conceptualization; Formal analysis; Investigation; Visualization; Methodology; Writing—original draft; Writing—review and editing; MG contributed to conceptualization, investigation, formal analysis and methodology for the majority of the data presented, she also made major contributions to visualization of the data and writing of the manuscript.

**Antonio Z Politi**: Formal analysis; Methodology; Writing—original draft; formal analysis, methodology and writing related to the mathematical model of mitotic entry. **Luisa Welp**: Methodology; methodology of phosphoproteomics. **Jasmin Jakobi**: Investigation. **Henning Urlaub**: Resources; Funding acquisition; resources and funding acquisition related to the proteomics experiments. **Peter Lenart**: Conceptualization; Supervision; Funding acquisition; Writing—original draft; Writing—review and editing; PL contributed to conceptualization of all experiments, funding acquisition, supervision, and writing of the manuscript.

Source data underlying figure panels in this paper may have individual authorship assigned. Where available, figure panel/source data authorship is listed in the following database record: biostudies:S-SCDT-10_1038-S44318-025-00400-9.

## Funding

## Disclosure and competing interests statement

The authors declare no competing interests.

# Expanded View Figures

**Figure EV1.   Detailed characterization of the delayed mitotic entry in plk1-inhibited cells.**

(**A**) HeLa cells expressing PCNA-mEGFP revealing S-phase progression after thymidine release. (**B**) Quantification of the S-phase duration in DMSO ($n = 20$) and BI 2536-treated ($n = 17$) cells. Unpaired Student's $t$ test with Welch's correction was used for statistical significance assessment. n.s. corresponds to $p > 0.05$ ($p = 0.4028$). (**C**) Quantification of the duration from S-phase termination to NEBD and with and without thymidine synchronization in HeLa Kyoto cells measured by PCNA labeling. Unpaired Student's $t$ test with Welch's correction between DMSO ($n = 15$) and BI2536-treated ($n = 15$) with added thymidine, and DMSO ($n = 30$) and BI 2536-treated ($n = 17$) without adding thymidine was used for statistical significance assessment. **** corresponds to $p < 0.0001$, with added thymidine $p = 7.98E{-}05$ and without added thymidine $p = 1.35E{-}08$. (**D**) Immunofluorescence images of HeLa Kyoto cells stained for γ-H2AX antibodies. Example images (left) and quantification of the number of γ-H2AX foci per nuclei (right) are shown. The DMSO ($n = 27$) and BI2536-treated ($n = 14$) cells were analyzed 8 and 17 h after thymidine release, respectively. Unpaired Student's $t$ test with Welch's correction was used for statistical significance assessment. n.s. corresponds to $p > 0.05$ ($p = 0.5467$). (**E**) Model network modified from Rata et al, 2018 to include a plk1 dependent term. (**F**) Immunoblot analysis of samples collected at different time points after thymidine release and DMSO/BI2536 treatment. The cells were split after shaking off into supernatant (mitotic) and attached (interphasic) fractions. The blots were imaged after treatment with fluorescently-labeled secondary antibodies specific to the primary antibodies shown. Cdk1 was used as a loading control. For (**B**), (**C**), and (**D**): significance testing was done using unpaired Student's $t$-test with Welch's correction. For (**A**) and (**D**): scale bars are 10 μm. (**G**) Bifurcation graph of the mathematical model. Shown is the steady state active [cdk1-cyclin B] ($[CycBCdk1]$) as function of [cdk2-cyclin A] activity ($[CycACdk2_{Tot}]$). Stable and unstable steady states are indicated by a solid and dashed line, respectively. When cdk2-cyclin A activity is above the saddle node (SN) bifurcation only the high cdk1 activity steady state remains.

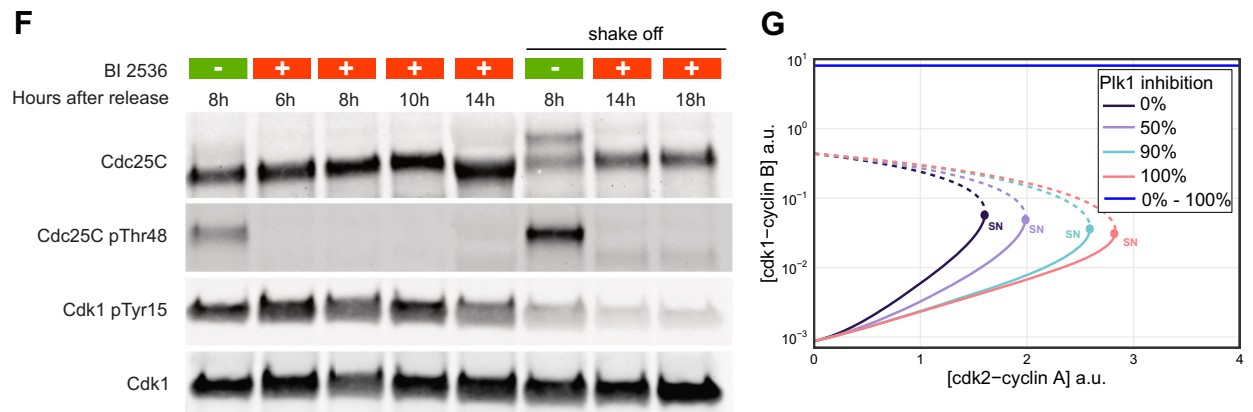

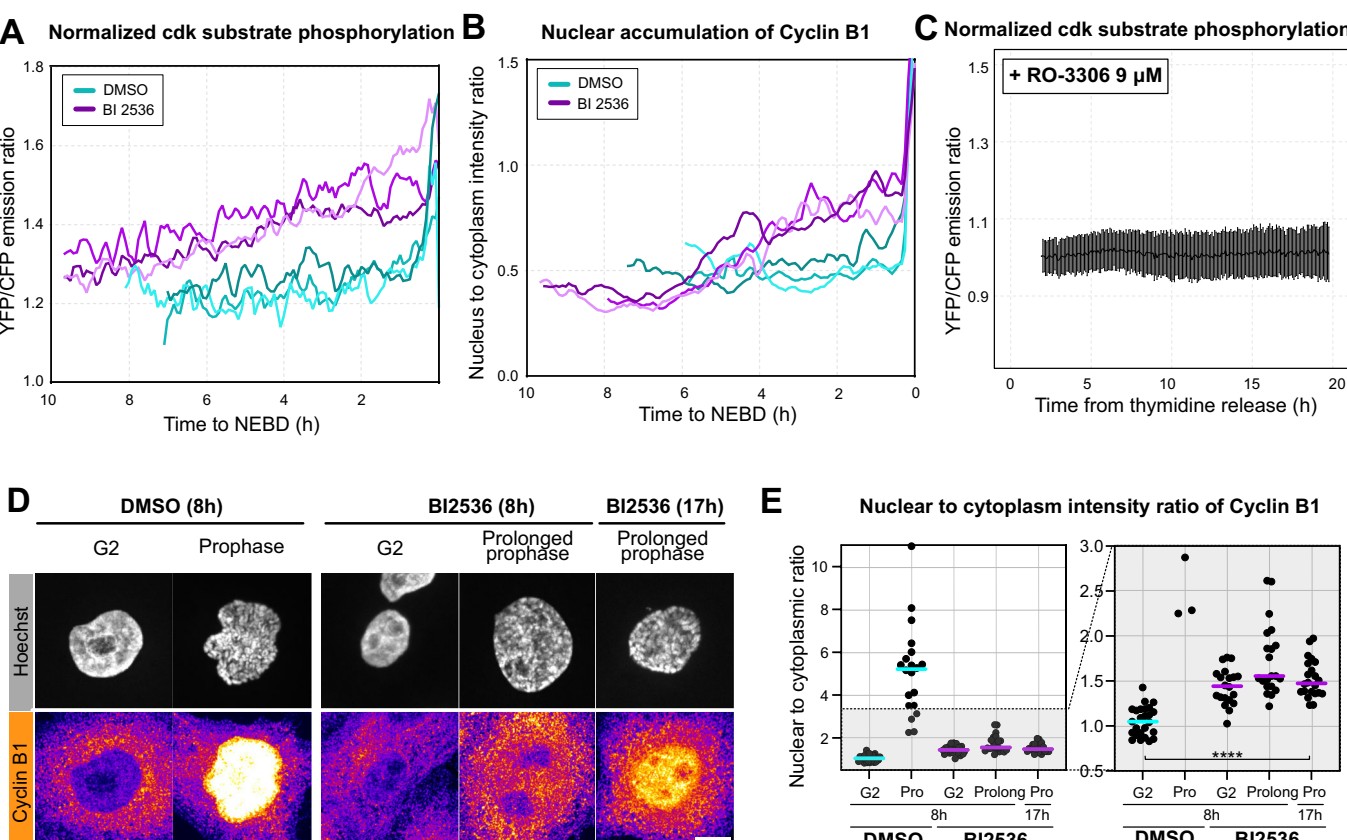

**Figure EV2.  Cdk1 activity and Cyclin B1 localization quantified in individual live cells and in fixed cells.**

(**A**) Quantification of FRET ratio changes of the Eevee-spCDK FRET probe in individual control and BI 2536-treated cells. (**B**) Quantification of Cyclin B1 nucleus to cytoplasm intensity ratio in individual control and BI 2536-treated cells. (**C**) Quantification of FRET ratio changes of the Eevee-spCDK FRET probe in RO 3306-treated population of cells ($n = 59$). Mean ± Standard Deviation (SD) is shown. (**D**) Immunofluorescence of cyclin B1 in control and BI 2536-treated cells fixed at different time points after thymidine release. Scale bar is 10 μm. (**E**) Quantification of nuclear to cytoplasmic ratio of fluorescence intensity of cyclin B1 in control (G2 ($n = 25$) and prophase ($n = 20$)), and treated cells (G2 ($n = 20$), prolonged prophase ($n = 22$) 8 h after thymidine release and prolonged prophase ($n = 22$) 17 h after thymidine release) on images similar to (**D**). Inset magnifying the lower range is shown on the right panel. Unpaired Student's $t$ test with Welch's correction was used for statistical significance assessment with **** corresponding to $p < 0.0001$ ($p = 2.22E{-}10$). Source data are available online for this figure.

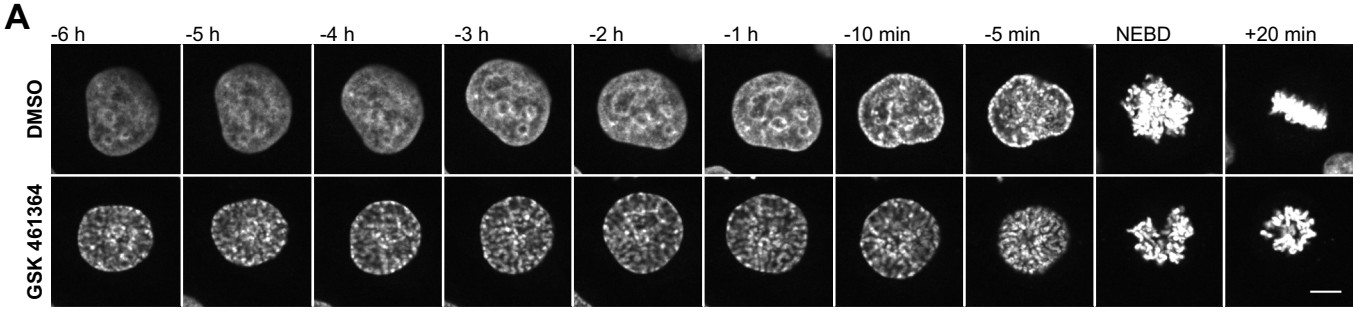

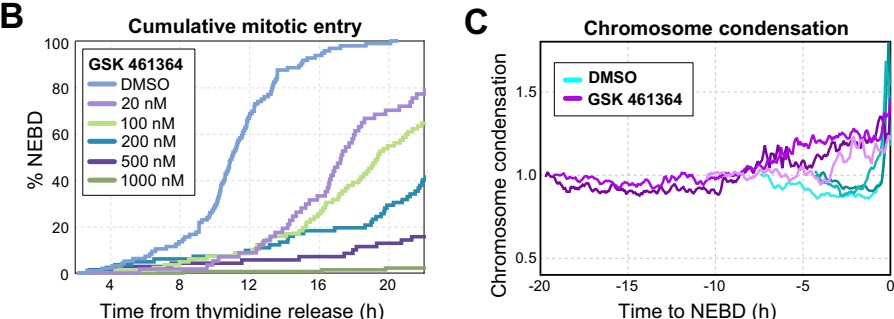

**Figure EV3.   A structurally unrelated Plk1 inhibitor recapitulates the phenotype caused by BI 2536.**

(A) Montages of HeLa cells showing mitotic entry of exemplary control and GSK 461364-treated cells stained with 5-SiR-Hoechst. Time is relative to NEBD. Scale bar is 10 μm. (B) Cumulative plots of NEBD timing in response to increasing concentrations of GSK 461364 in HeLa Kyoto cells. (C) Quantification of chromosome condensation by standard deviation of fluorescence intensity of individual HeLa Kyoto cells on recording similar to shown in (A).

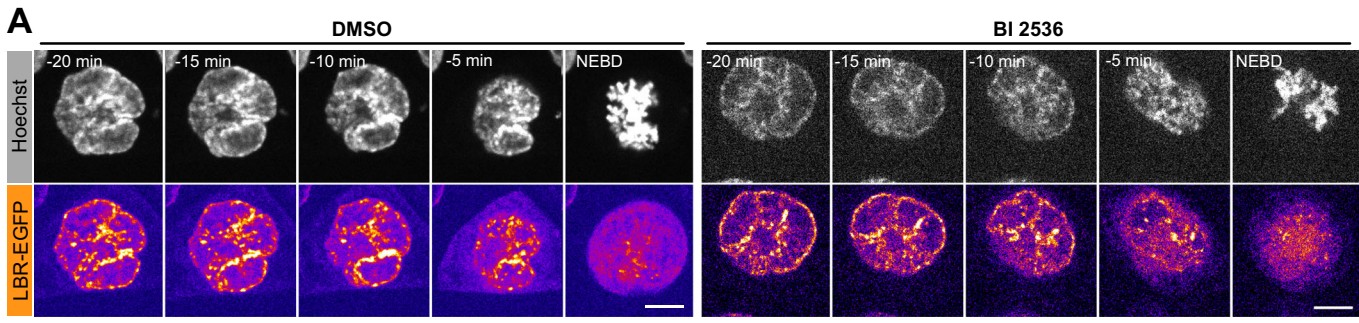

**B**   **Cytoplasmic LBR-EGFP intensity**

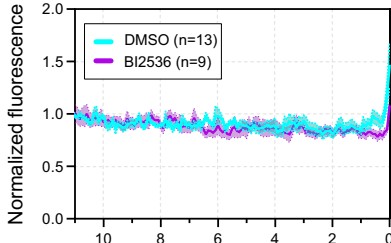

**Figure EV4. Release kinetics of LBR, a protein of the nuclear envelope, is unaffected by Plk1 inhibition.**

(A) Selected frames from a time lapse showing the localization of LBR-EGFP in the time leading to NEBD in DMSO and BI 2536-treated cells. Scale bar is 10 µm. (B) Quantification of LBR-EGFP mean cytoplasmic fluorescence intensity in DMSO and BI 2536-treated cells on recordings similar to (A). Data are normalized to the mean intensity in the first 5 frames of imaging. Mean ± standard error of mean (SEM) is shown.

## A

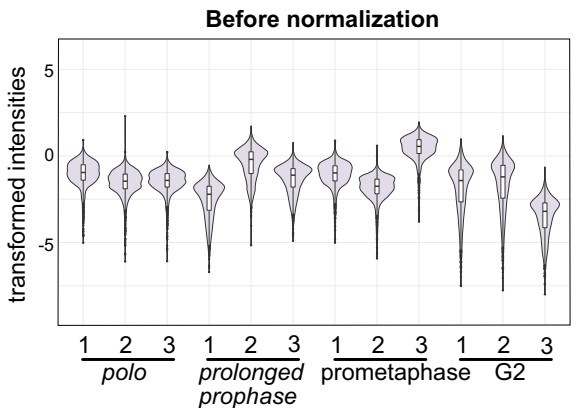

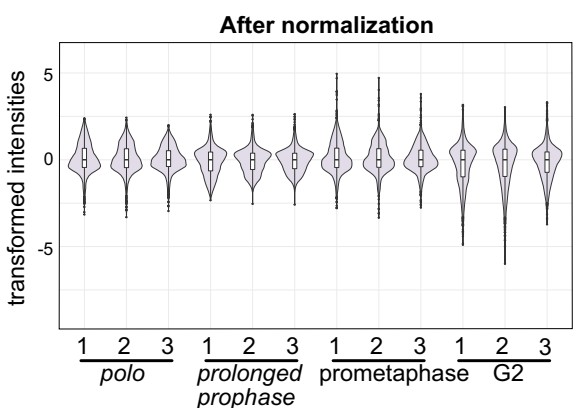

## B Differentially phosphorylated proteins between *polo* and Prometaphase samples

| CENPF***1121 |
| BABAM1***65 |
| NOLC1***643 |
| TPR***2048 |
| NUP98***1023 |
| CEP97***308 |
| NCAPH*15 |
| PARN***557 |
| MISP***541 |
| TMPO***168 |
| CEP170**630 |
| HNRNPU*59 |
| ANLN*225 |
| PALLD**893 |
| AKAP8**112 |
| MDC1*966 |
| MKI67*1376 |
| CLASP1*646 |
| BABAM1**29 |
| KIF21A**1231 |
| SPECC1L*384 |
| LMNA**277 |
| TJP3*856 |
| LMNA**652 |
| CENPF*144 |
| NUP98**623 |
| TPR*2037 |
| ANLN**54 |
| CEP170***785 |
| NUP98**670 |
| CDKN1A***130 |

Log fold change

2.5

0.0

−2.5

## C Kinases negatively enriched in *polo* sample relative to prometaphase

| Kinase | Kinase Group | Log2 Enrichment | Significance (-log adjusted p value) |
|---|---|---|---|
| PLK1 | Other | -2.9989 | 8.884269 |
| CAMK2D | CAMK | -2.05404 | 5.461809 |
| PLK4 | Other | -2.23991 | 2.778463 |
| MAPKAPK3 | CAMK | -2.06998 | 2.547787 |
| MAPKAPK2 | CAMK | -1.55397 | 2.547787 |
| CHK1 | CAMK | -1.81409 | 2.524474 |
| GCN2 | Other | -1.91386 | 2.524474 |
| DLK | TKL | -2.1509 | 2.334321 |
| PKN3 | AGC | -2.44636 | 2.302078 |
| CAMK4 | CAMK | -2.61347 | 2.247387 |

## D Kinases enriched in prolonged prophase

| Kinase | Kinase Group | Log2 Enrichment | Significance (-log adjusted p value) |
|---|---|---|---|
| KIS | Other | 0.924796 | 4.592543 |
| CDK5 | CMGC | 0.719101 | 2.568039 |
| CDK2 | CMGC | 0.645396 | 2.357501 |
| JNK3 | CMGC | 0.732357 | 2.357501 |
| CDK8 | CMGC | 0.660207 | 2.234585 |
| CDK13 | CMGC | 0.643014 | 1.462511 |
| NLK | CMGC | 0.698971 | 1.362713 |
| CDK12 | CMGC | 0.637616 | 1.362713 |
| CDK16 | CMGC | 0.557815 | 1.362713 |
| CDK1 | CMGC | 0.5435 | 1.362713 |

◀  **Figure EV5.   Kinase profiling confirms specific inhibition of plk1 by BI 2536, and cdks as the primary kinases active in the prolonged prophase state.**

(**A**) Violin plots showing the distribution of the log(2) transformed intensities of the phosphosites detected in the 12 samples (4 samples in 3 replicates) before and after normalization. For the shown box plots, the center corresponds to the median and the lower and upper hinges correspond to the first and third quartiles (the 25th and 75th percentiles). The upper and lower whisker extend from the hinge to the largest and smallest value, respectively, no further than 1.5 * IQR from the hinge (where IQR is the inter-quartile range, or distance between the first and third quartiles). Data beyond the end of the whiskers are outliers and are plotted individually. (**B**) Differentially phosphorylated peptides in *polo* cells in comparison to prometaphase cells that were detected using linear modeling. The gene names of the corresponding peptides are shown, the asterisks mark the adjusted *p*-value with ***, **, * corresponding to $p \leq 0.001$, 0.01, and 0.05, respectively, followed by the position of the phosphosite within the amino acid sequence of the protein. The color coding corresponds to the log(2) fold change of intensity values. Statistical significance was assessed using a moderated *t*-test where empirical Bayes moderation was applied to stabilize variance estimates across genes. (**C**) List of kinases least enriched in the polo sample in comparison to the prometaphase sample after applying linear modeling. (**D**) List of kinases most enriched in the prolonged prophase sample after applying linear modeling. For (**C**) and (**D**), the enrichments were determined using one-sided exact Fisher's tests and corrected for multiple hypotheses using the Benjamini–Hochberg method.

