## [Peer Review File · The EMBO Journal]

PLK1 inhibition delays mitotic entry revealing changes to the phosphoproteome of mammalian cells early in division

Monica Gobran, Antonio Politi, Luisa Welp, Jasmin Jakobi, Henning Urlaub, and Peter Lenart

Corresponding author(s): Peter Lenart (plenart@mpibpc.mpg.de)

Review Timeline:

Submission Date:	28th May 24
Editorial Decision:	11th Jul 24
Revision Received:	28th Nov 24
Editorial Decision:	17th Jan 25
Revision Received:	3rd Feb 25
Accepted:	4th Feb 25

Editor: Ioannis Papaioannou

Transaction Report:

Dear Dr. Lenart,

Thank you for submitting your manuscript EMBOJ-2024-118013 for consideration by The EMBO Journal and for your patience during peer review. Your manuscript has now been seen by three experts in the field, and we have received the full set of their comments, which you can find below.

As you will see, all referees recognize the importance of the topic as well as the high quality of the experimental investigation and the presented data. Referees #1 and #2 further point out that this work makes a significant contribution to a more comprehensive understanding of the processes that precede mitosis, while the proteomic analysis provides the cell cycle community with a useful resource. A common criticism raised by referees #2 and #3, on the other hand, is that the work is mostly descriptive, while referee #3 also finds the conceptual advance over the previous literature incremental.

On balance, and upon discussion of the well-informed and constructive referee reports in our editorial team, we are open to considering a revised version of your manuscript for publication in The EMBO Journal should you be willing and able to embark on a major revision that would sufficiently strengthen the manuscript addressing the relevant and useful suggestions that all three referees list in their reports. In particular, I would like to emphasize that we find the major conceptual concern raised by referee #2 (regarding the very nature of the studied "prolonged prophase") particularly relevant, and I would like to encourage you to consider and address it carefully during your revision. Furthermore, we agree with referee #3 that it would be informative to include some data on the phosphorylation status of CDC25C and Cdk1.

Please include in your resubmission a detailed point-by-point response addressing all referees' comments. I should add that it is EMBO Journal policy to allow only a single round of major revision, and acceptance of your manuscript will therefore depend on the completeness of your responses in this revised version. Given the nature of the referee concerns and comments, I would encourage you to share with me a revision plan and a draft response to the major points made by the referees already at the early stages of your revision, so that we can discuss further the extent and feasibility of the additional requested experiments.

We generally allow three months as standard revision time (October 10, 2024). As a matter of policy, competing manuscripts published during this period will not negatively impact our assessment of the conceptual advance presented by your study. However, we request that you contact us as soon as possible upon publication of any related work, to discuss how to proceed. Should you foresee a problem in meeting this three-month deadline, please let us know in advance and we may be able to grant an extension. Please also let me know if you have any questions or comments that you would like to discuss with me.

Thank you for the opportunity to consider your work for publication in The EMBO Journal. I look forward to your revision plan and point-by-point response to the referee comments at your earliest convenience.

Best regards,

Ioannis

Instructions for preparing your revised manuscript

1. When you are ready to submit the revision, please upload:

- A Word file of the manuscript text (including legends of main Figures, EV Figures and Tables). Please make sure that changes are highlighted (or "tracked") to be clearly visible.

- Individual production-quality figure files (one file per figure). When assembling your figures, please refer to our figure preparation guidelines in order to ensure proper formatting and readability in print as well as on screen:

If the data shown in a figure are obtained from n {less than or equal to} 2, please use scatter plots showing the individual data points.

- i. the name of the statistical test used to generate error bars and P values
- ii. the number (n) of independent experiments (please specify technical or biological replicates) underlying each data point (discussion of statistical methodology can be reported in the Materials and Methods section, but figure legends should contain a basic description of n, P, and the test applied)
- iii. the nature of the bars and error bars (s.d., s.e.m.).

- A point-by-point response to the referees' comments, with a detailed description of the changes made (as a word file). All referees' concerns must be fully addressed and their suggestions taken on board. When preparing your letter of response to the referees' comments, please bear in mind that this will form part of the Review Process File and will therefore be available online to the community. Please note that you have the possibility to opt out of the transparent process at any stage prior to publication by letting the editorial office know (contact@embojournal.org); if you do opt out, the Review Process File link will point to the following statement: "No Peer Review File is available with this article, as the authors have chosen not to make the review process public in this case.". For more details on our Transparent Editorial Process, please visit our website: <https://www.embopress.org/page/journal/14602075/authorguide#transparentprocess>

- Expanded View (EV) files (replacing Supplementary Information) that are collapsible/expandable online. A maximum of 5 EV Figures can be typeset. EV Figures should be cited as "Figure EV1, Figure EV2" etc. in the text, and their respective legends should be included in the manuscript file after the legends of regular figures. See detailed instructions regarding Expanded View files here:

- For the figures that you do NOT wish to display as Expanded View figures, they should be bundled together with their legends in a single PDF file called "Appendix", which should start with a short Table of Contents (including page numbers). Appendix figures should be referred to in the main text as: "Appendix Figure S1, Appendix Figure S2" etc. Please see detailed instructions here: <https://www.embopress.org/page/journal/14602075/authorguide#expandedview>

- A complete author checklist, which you can download from our author guidelines (<https://www.embopress.org/page/journal/14602075/authorguide>). Please note that the checklist will also be part of the Review Process File.

2. Please note that no statistics should be calculated and shown in Figures if n=2. Please also note that each p value should be reported as an exact value.

3. Before submitting your revision, primary datasets (and computer code, where appropriate) produced in this study need to be deposited in appropriate public databases (see <https://www.embopress.org/page/journal/14602075/authorguide#dataavailability>).

In particular, you are kindly requested to deposit the mass spectrometry data that were generated in your study in an appropriate repository. The accession number, database, and the specific URL (link) should be listed in a formal "Data availability" section (placed after Materials and Methods) that follows the model below (see also <https://www.embopress.org/page/journal/14602075/authorguide#dataavailability>):

Data availability

- RNA-seq data: Gene Expression Omnibus GSE46843 (<https://www.ncbi.nlm.nih.gov/geo/query/acc.cgi?acc=GSE46843>)
- [data type]: [name of the resource] [accession number/identifier/doi] ([URL or identifiers.org/DATABASE:ACCESSION])

*** All links should resolve to a page where the data can be accessed. ***

*** Please remember to provide in the Data availability section of your revised manuscript reviewer passwords if the datasets are not yet public. ***

*** The Data Availability Section is restricted to new primary data that are part of this study. In case you have no data that require deposition in a public database, please state so instead of referring to the database: "Our study includes no data deposited in public repositories." under the heading "Data availability". ***

4. Please check that the title and the abstract of the manuscript are brief, yet explicit, even to non-specialists. The length of the title should not exceed 100 characters, and the abstract should be a single paragraph not exceeding 175 words.

5. The Materials and Methods need to be described in the manuscript using our "Structured Methods" format, which is now required for all research articles. According to this format, the Materials and Methods section includes a single "Reagents and Tools Table" -listing key reagents, experimental models, software and relevant equipment and including their sources and

relevant identifiers- followed by a "Methods and Protocols" section describing the methods. More information on this format as well as detailed instructions, examples, and a template (.docx) for the "Reagents and Tools Table" can be found in our author guide: <https://www.embopress.org/page/journal/14602075/authorguide#structuredmethods>.

6. Please also note our reference format: <https://www.embopress.org/page/journal/14602075/authorguide#referencesformat>.

8. Please remember: digital image enhancement is acceptable practice, as long as it accurately represents the original data and conforms to community standards. If a figure has been subjected to significant electronic manipulation, this must be noted in the figure legend or in the "Materials and Methods" section. The editors reserve the right to request original versions of figures and the original images that were used to assemble the figure.

9. Our journal encourages inclusion of data citations in the reference list to directly cite datasets that were obtained from public databases. Data citations in the article text are distinct from normal bibliographical citations and should directly link to the database records from which the data can be accessed. In the main text, data citations are formatted as follows: "Data ref: Smith et al, 2001" or "Data ref: NCBI Sequence Read Archive PRJNA342805, 2017". In the Reference list, data citations must be labeled with "[DATASET]". A data reference must provide the database name, accession number/identifiers, and a resolvable link to the landing page from which the data can be accessed at the end of the reference. Further instructions are available at: <https://www.embopress.org/page/journal/14602075/authorguide#referencesformat>.

10. We request authors to consider both actual and perceived competing interests. Please review our policy (<https://www.embopress.org/page/journal/14602075/authorguide#conflictsofinterest>) and update your competing interests statement if necessary. Please name this section 'Disclosure and competing interests statement' and place it after the Acknowledgements section.

11. Please note that all corresponding authors are required to provide an ORCID ID upon submission of a revised manuscript (<https://orcid.org/>). Please find instructions on how to link your ORCID ID to your account in our manuscript tracking system in our Author guidelines (<https://www.embopress.org/page/journal/14602075/authorguide#authorshipguidelines>).

12. We use CRediT to specify the contributions of each author in the journal submission system. CRediT replaces the author contribution section, which should be removed from the manuscript. Please use the free text box to provide more detailed descriptions. See also guide to authors: <https://www.embopress.org/page/journal/14602075/authorguide#authorshipguidelines>.

14. We would also welcome the submission of cover suggestions or motifs to be used by our Graphics Illustrator in designing a cover.

15. Please use the link below to submit your revision:
<https://emboj.msubmit.net/cgi-bin/main.plex>

Referee #1:

The manuscript „Plk1 inhibition delays mitotic entry revealing prophase-specific changes to the phosphoproteome" by Peter Lenart and colleagues show that plk1 inhibition causes a delay in mitotic entry in a prophase-like state that displays progressively condensing chromosomes, increased microtubule dynamics, reorganization of the actin cortex, while the nuclear envelope remains intact. The following phospho-proteomic analysis reveals a large number of mitotic phosphorylations, which presents a novel dataset of kinase target sites that may be directly involved in regulation of cell cycle progression from G2 to prophase-prometaphase and metaphase. Indeed, the phospho-regulation of cell cycle progression in prophase has been still relatively poorly understood. The study is technically very elegant and is certainly a very important step forward promoting our detailed understanding of processes that lead from interphase to mitosis. Before publishing, the authors should address few points outlined below to make the paper and its concepts more clear and understandable. The parallel drawn from the Novak and Hochegger bistability paper needs to be better explained, and also, the

seeming controversy of plk1 as a potential activator of Cdc25 and Cdk1: why it delays the Cdk1-CycB activity in unperturbed cycle while causes premature and progressive Cdk1 activity accumulation during the prolonged prophase in the inhibition experiment? After these concerns are addressed, I would be very enthusiastic to recommend the manuscript for publishing.

1. The study demonstrates that upon plk1 inhibition the cdk1 activity increases gradually over several hours with individual cells stochastically reaching the mitotic entry threshold. Based on this and other experimental observations, the authors conclude that plk1 acts as an effective catalyst of prometaphase transition, ensuring that NEBD occurs robustly and within a narrow time window. However, they also mention one major pathway (Gheghiani et al., 2017) identified to activate cdc25 as follows: cdk1/cyclin A phosphorylates the protein Bora, leading to the activation of Aurora A, which in turn phosphorylates and activates plk1. Plk1 then phosphorylates and activates cdc25C, which then dephosphorylates and activates cdk1. The authors should explain how inhibiting the plk1 activating function towards cdk1 will, in fact, prematurely activate Cdk1 during the prolonged prophase, albeit in a slower and progressively accumulating mode, but still, the plk1 inhibition seemingly activates cdk1. Could there be another branch of plk1 pathway in prophase that delays the cdk1 activity? Could the authors predict it from the proteomics data?
2. A scheme, or a wiring diagram, analogous to the one presented in Novak & Hochegger paper (Figure 3) would help the general reader a lot. It could explain how the authors envision the regulatory connections among the regulators plk1, cdk1, wee1, cdc25, mitotic substrate pool, etc.
3. Similar concerns as in Point 1 above apply to the following interpretation: "In normal prophase, these early changes to the nuclear envelope have been proposed to facilitate nuclear accumulation of cdk1-cyclin B (Gavet and Pines, 2010), which is then key to full activation of cdk1, driving NEBD and transition to prometaphase (Santos et al., 2012). Plk1 inhibition may prevent these changes in nuclear transport causing the observed slow and limited relocalization of cyclin B to the nucleus and thus slow and gradual increase in cdk1 activity." How does the plk1 inhibition in Figure 4 cause both accumulation of Cyclin B and cdk1 activity: "cdk activity detected by the FRET probe was almost identically mirrored by slow and gradual relocalization of cyclin B1 to the nucleus?"
4. How do the authors interpret relationship between the slow accumulation of cdk1 activity (plk1 inhibition) vs abrupt switch (no inhibition) and the progression of phosphorylation dependent events in prophase? It is unclear how these rare "stochastic" mitotic entries in plk1 inhibitor experiment can occur in case of slow cdk1 accumulation, which would likely mean that only an occasional incremental change can trigger the mitosis in these stochastic events, while in normal cell cycle one needs a switchlike input of cdk1. Also, as the cyclin expression levels are very variable and overlapping when populations of individual cells are analyzed (Figure 4 E,F), how can the tight and abrupt switch be explained by a cdk1 threshold? These thresholds must be really flexible?
5. In connection with the previous, and the model recently published by the Hochegger and Novak groups (Rata et al., 2018) suggesting that mitotic entry relies on two inter-linked bistable switches, the positive feedback loop activating cdk1, and the feedback through Greatwall leading to the inactivation of the PP2A:B55 phosphatase. However, Hochegger and Novak did not integrate the plk1 loop into their model and wiring diagram. Please explain how the plk1 enters into the diagram, and most importantly, how the non-stable steady state that was predicted by modeling and experimentally demonstrated in Hochegger and Novak paper by double inhibition of Wee1 and Greatwall is analogous, if not the same intermediate state, that the authors show in the current paper, as a result of plk1 inhibition.
6. If the activation is cdc25 phosphorylation is dependent on plk1, then accumulation of Cyclin B and inhibitory phospho-Tyr on cdk1 should not be in correlation (when plk1 is inhibited). Or at least, higher plk1 inhibition doses should cause more phospho-Tyrosine signal. Can this be tested quantitatively, similarly as in Fig 4 D and E in the Hochegger paper?
7. The authors could perhaps tone down the following statement: "Prolonged prophase thus serves as a model for an otherwise inaccessible transient stage of mitosis, early, before cdk1 is fully activated." It is still a very much different situation as the inhibitor treatment changes a lot of signaling dynamics or likely hundreds of kinase targets compared to the normal prophase. "Thus, plk1 inhibition may interfere with the positive feedback loop activating cdk1, while the PP2A:B55 branch remains intact. In this context, prolonged prophase may be an ideal condition to analyze the complex signaling network regulating mitotic entry." Similar concern: in case cdk1 activity slowly increases and the phosphatase is not affected, then the phosphorylation stoichiometries hardly resemble prophase situation and cannot be called as an "ideal condition".

Referee #2:

This manuscript by Gobran and colleagues reports the effects of PLK1 inhibition on the timing of mitotic entry and mitotic progression. These effects have been studied before, including in a seminal 2007 paper from the manuscript's senior author. In this new study, the authors used specific PLK1 inhibitors to verify two fundamental reported effects of PLK1: 1) a (usually time-limited) arrest in mitotic prophase and 2) an arrest in prometaphase with a "polo" phenotype (essentially a monopolar spindle). After characterizing the prophase arrest in detail, the authors came to the conclusion that it represents a special state, at least in part distinct from G2, and performed a phosphoproteomic analysis to identify phosphorylation sites characteristic of this phase.

Briefly, in Figure 1 the authors use three distinct human cell lines to determine that PLK1 inhibition (with the specific and selective inhibitor BI2536) delays mitotic entry and, after mitotic entry, causes cells to display a polo phenotype. In Figure 2 they demonstrate premature chromosome condensation long before mitotic entry. In Figure 3, they demonstrate that several aspects of cell physiology, including microtubule and actin dynamics, but not nuclear transport, are perturbed during a prolonged prophase. In Figure 4, they use a CDK FRET sensor to investigate the timing of CDK mitotic activation, and demonstrate that

CDK activity grows progressively in PLK1-inhibited cells, rather than abruptly as in control cells. In Figures 5 and 6 the authors present the results of a phosphoproteomics analysis that identified various phosphorylation sites in G2, prophase-like, and prometaphase states. An important conclusion of this analysis is that phosphorylation status in the prolonged-prophase state induced by PLK1 inhibition is more similar to that in G2 cells than to that in prometaphase cells, although differences were also measured.

Overall, the study is rigorous and technically very well done, and the controls are adequate. The conclusions consolidate and extend previous, more scattered evidence. The proteomic analysis provides an important new tool for researchers interested in the mechanisms of cell cycle control. On the weak side 1) there is a significant conceptual concern, as explained below; 2) the study remains somewhat superficial on the mechanistic level, although this may be expected given that this is an already significant contribution. In this context, a more effective presentation of the results of the proteomics analysis would give the manuscript a more mechanistic touch that is currently missing.

Specific points

- Prophase is intrinsically transient. The authors may have identified a condition to prolong prophase, but this condition is achieved by eliminating PLK1, which is a crucial component of the switch and itself also a kinase targeting many different substrates. This is clearly different from activating a checkpoint to block cells in a certain cell cycle regime without touching the relevant cell cycle kinases (of note, there is no prophase checkpoint that we know of). Chromosome condensation and the dotted pH3 S10 signal are the only hallmark of prophase observed by the authors. Other observed aspects, e.g. microtubule plus-tip velocities, do not fit the G2 or the M-phase pattern, being even higher than in mitosis. Thus, I am doubtful that the state studied by the authors is, strictly speaking, a prolonged prophase, at least not in the way that we consider a prolonged mitosis the arrest obtained with a spindle poison, for instance. A possible way to address this conundrum would be to extend the proteomics analysis to include a comparison with cells arrested before mitosis through inhibition of CDK1 with RO3306, a condition that the authors, in the Discussion, equate to prophase.

-From section 2 of the results onwards, the authors perform all their experiment in HeLa cells. It would be useful to state so in the main text, and clarify this choice for the readership.

-Figure 4C: CDK1 FRET signal is less steep but higher in the BI condition. Can the authors comment on this?

-Figure 5A: Why not compare BI+STLC at 5h and 13.5h? Wouldn't STLC at 5h, BI+ STLC at 5h, STLC at 13.5h, BI + STLC at 13.5h have been a better control?

-The important proteomics data in figures 5 and 6 is presented rather sketchily. Readers can only access the critical information by referring to the tables, which is time-consuming. Which enriched or depleted phosphosites in the prolonged prophase group are actually specific prophase phosphorylations? Are the identified sites merely more or less abundant "copies" of G2 or prometaphase sites? Figure S4B is nice and could be placed in the main figures, extended to include the prophase enriched sites. This would be more informative, as currently the identified sites are classified based on the function of the phosphorylation protein, and any information on the progression from G2 to prophase to mitosis is missing. In addition to the GO classification, it would be important to annotate which of the identified proteins are known to be direct PLK1 substrates (based on previous literature) and whether the identified sites conform to a consensus. Also, how many sites are in common between G2 and the ProPro state, how many between ProPro and M, etc.? This could be easily presented with the classical "overlapping circle" plots.

Minor points

-I would recommend removing adverbs of subjective surprise and interest, from "strikingly" to "intriguingly". The last word on this is with the editors, though.

-Introduction, end of first paragraph: the authors should clarify that it is possible to arrest cells in G2 with RO3306, and explain in what way this is different from what they want to do. This is all the more important in view that in the Discussion the authors claim that the prolonged prophase arrest they have obtained by inhibiting PLK1 phenocopies CDK1 inhibition.

-Results, first paragraph: "...compare side-by-side three cell lines..."

-Results, second paragraph: "...we escalated doses to multiples of the concentrations shown to suffice..."

-The authors show that S-phase progression is not affected by PLK1 inhibition (Figure S1A-C). The authors may consider including a marker for G2-M progression to show that also G2 is not affected, and specifically by monitoring the disappearance of PCNA speckles.

-Figure 3D would be more intuitive to read if the authors explained directly in the main text what the magenta arrow means.

- Page 5, last paragraph: "regulatory kinases"; which ones?
- Throughout: "data" is plural, i.e. data are, not is
- Page 7, third paragraph: "...harboring phosphosites of different abundance"
- Page 8, elimination of CENP-F has absolutely mild effects on kinetochore assembly and the paper referred to demonstrated a role in Dynein-Dynactin recruitment or activation at the kinetochore.
- Figures 5 and 6 (including legend): what does "specific to prolonged prophase" mean exactly? What is being shown exactly in the volcano plot in Figure 5? Is it a comparison with G2? I could not tell what exactly is displayed.
- Figure 2E: Two different scales are used, but it seems the percentages of dotted cells after release are quite different in the control and BI-treated samples. Any hint why that is the case?
- Discussion, first paragraph: "somewhat contentious". Relevant references have to be added to support the statement.
- In Figure S2, Nup107 signal looks less clear of LBR and it seems more intense on the chromosomes at NEBD in the BI condition. Can this signal be quantified as well? If not, it may be better to remove it.
- Figure S3C should be called as S3A as it comes first in the main text.
- It would be nice to include the number of cells in each graph where the population average behavior was plotted.
- The authors should consider including statistical analysis to Figure 3B.
- In the beginning of section 4, the opening sentence "Thus, prolonged prophase ... and microtubule cytoskeleton" sounds like the conclusion of a paragraph rather than the beginning of a new one. I would suggest rephrasing it for clarity.
- The authors could spend more time discussing interesting potential targets. For instance, do they detect differential phosphorylation by the Cdc25C1 phosphatase in their prolonged prophase sample? This would make their point stronger.
- In section 6 of the results, the authors show that NCAPH2 and NCAPD3 are differentially phosphorylated in the prolonged prophase sample. Since it is now known that M18BP1 phosphorylation by CDK1 recruits Condensin II to chromosomes during mitotic entry, it would be nice to also show whether M18BP1 is detected in the screen.

Referee #3:

Plk1 is a key cell cycle kinase which has important roles in regulating mitosis at different stages. One critical role for Plk1 is the control of mitotic entry, and it had already been observed 20 years ago that RNAi-mediated depletion of Plk1 delays mitotic entry, an observation that was subsequently confirmed by small molecule inhibition of Plk1 (Sumara et al., 2004; Lenart et al., 2007). The effect of delaying entry into mitosis and prolonging prophase is most likely due decreased activating phosphorylation of the CDC25C phosphatase by Plk1, resulting in delayed removal of the inhibitory Thr14/Tyr15 phosphorylations, and it has been previously shown that phospho-mimetic versions of CDC25C largely rescue the mitotic entry defect in cells treated with Plk1 inhibitor (Gheghiani et al., 2017).

In this manuscript Lenart and colleagues use live cell imaging to carefully analyse the mitotic entry delay in different cell lines, and characterise the effects on chromosome condensation, the microtubule cytoskeleton and cyclin B localisation. They come to the conclusion that Plk1 inhibition results in a prolonged prophase, and then aim to exploit this observation by performing a mass spec analysis of these cells in comparison with mock-treated cells to identify phosphorylations that characterise prophase.

This is a nicely conducted manuscript with high-quality data. Unfortunately, though, the study is almost entirely descriptive and provides very little novel insight into either the role of Plk1 in regulating mitotic entry or the biology of prophase. Surprisingly, the supposedly key effector of Plk1, the phosphatase CDC25C is not analysed in the manuscript. It would be informative to know whether the phosphorylation status of CDC25C and its substrate Cdk1 change in accordance with expectations and whether all the changes that Gobran et al. observe are due to insufficient CDC25C activation. Furthermore, it would be interesting to know whether the Plk1 inhibition phenotypes could be rescued by expressing a phospho-mimetic version of CDC25C, although this line of experimentation is very close to the experiments that have already been published by Gheghiani et al., 2017.

The proteomic analysis of the prolonged prophase state found in Plk1-inhibitor treated cells, could in principle be interesting but in its current form, although the analysis of the data identifies proteins that are differentially phosphorylated in the Plk1-inhibitor-treated cells, there is no identification of the kinases that normally carries out these phosphorylations. This should most likely be

Cdk1-cyclin B1 but it would be good to have this confirmed.

Can the authors identify any key substrates that would push cells into mitosis if a phospho-mimetic mutant was expressed?

Altogether, this is a nice confirmation of the role of Plk1 in promoting mitotic entry but there is little advance over already published studies.

Minor comments:

-Table S3 with the list of differentially phosphorylated proteins in prophase was not included with the submitted manuscript, hence an evaluation of these data was not possible.

-*"We also identified phosphorylation of the TTK/MPS1 kinase, which regulates the assembly of the mitotic checkpoint complex on the kinetochore (Schweizer et al., 2013)."*

"Schweizer et al., 2013" is not the correct reference for this statement. Mitotic phosphorylation of human MPS1 has been demonstrated by Diril et al., 2016, Plos Genet.; and Hayward et al., JCB, 2019, and these two publications would be better references for this statement.

Dear Dr. Lenart,

Thank you for submitting your manuscript EMBOJ-2024-118013 for consideration by The EMBO Journal and for your patience during peer review. Your manuscript has now been seen by three experts in the field, and we have received the full set of their comments, which you can find below.

As you will see, all referees recognize the importance of the topic as well as the high quality of the experimental investigation and the presented data. Referees #1 and #2 further point out that this work makes a significant contribution to a more comprehensive understanding of the processes that precede mitosis, while the proteomic analysis provides the cell cycle community with a useful resource. A common criticism raised by referees #2 and #3, on the other hand, is that the work is mostly descriptive, while referee #3 also finds the conceptual advance over the previous literature incremental.

On balance, and upon discussion of the well-informed and constructive referee reports in our editorial team, we are open to considering a revised version of your manuscript for publication in The EMBO Journal should you be willing and able to embark on a major revision that would sufficiently strengthen the manuscript addressing the relevant and useful suggestions that all three referees list in their reports. In particular, I would like to emphasize that we find the major conceptual concern raised by referee #2 (regarding the very nature of the studied "prolonged prophase") particularly relevant, and I would like to encourage you to consider and address it carefully during your revision. Furthermore, we agree with referee #3 that it would be informative to include some data on the phosphorylation status of CDC25C and Cdk1.

Please include in your resubmission a detailed point-by-point response addressing all referees' comments. I should add that it is EMBO Journal policy to allow only a single round of major revision, and acceptance of your manuscript will therefore depend on the completeness of your responses in this revised version. Given the nature of the referee concerns and comments, I would encourage you to share with me a revision plan and a draft response to the major points made by the referees already at the early stages of your revision, so that we can discuss further the extent and feasibility of the additional requested experiments.

We generally allow three months as standard revision time (October 10, 2024). As a matter of policy, competing manuscripts published during this period will not negatively impact our assessment of the conceptual advance presented by your study. However, we request that you contact us as soon as possible upon publication of any related work, to discuss how to proceed. Should you foresee a problem in meeting this three-month deadline, please let us know in advance and we may be able to grant an extension. Please also let me know if you have any questions or comments that you would like to discuss with me.

Thank you for the opportunity to consider your work for publication in The EMBO Journal. I look forward to your revision plan and point-by-point response to the referee comments at your earliest convenience.

Best regards,
Ioannis

Dear Ioannis,

Please find below our detailed response to the reviewers. In brief, following the suggestion of reviewer 1, we incorporated in the manuscript a previously published computational model of the signaling network driving mitotic entry (Rata et al., 2018). We simulated the effects of plk1 inhibition that, to our delight, readily recapitulated key features observed in experiments. This greatly helped us to clarify how plk1 affects mitotic entry, precisely addressing the conceptual concern raised by reviewers 1 and 2. Incorporating these new findings required a relatively major editing of the manuscript text, but we believe they improved it greatly. Additionally, we performed Western blots to address the concerns of reviewer 3 showing the phosphorylation status of cdc25c and cdk1 in plk1 inhibited cells. In addition, we improved and extended analyses of the proteomics data as requested by reviewers 2 and 3.

Best regards,
Peter

Response to the reviewers

Referee #1

The manuscript "Plk1 inhibition delays mitotic entry revealing prophase-specific changes to the phosphoproteome" by Peter Lenart and colleagues show that plk1 inhibition causes a delay in mitotic entry in a prophase-like state that displays progressively condensing chromosomes, increased microtubule dynamics, reorganization of the actin cortex, while the nuclear envelope remains intact. The following phosphoproteomic analysis reveals a large number of mitotic phosphorylations, which presents a novel dataset of kinase target sites that may be directly involved in regulation of cell cycle progression from G2 to prophase-prometaphase and metaphase. Indeed, the phospho-regulation of cell cycle progression in prophase has been still relatively poorly understood. The study is technically very elegant and is certainly a very important step forward promoting our detailed understanding of processes that lead from interphase to mitosis.

Before publishing, the authors should address a few points outlined below to make the paper and its concepts more clear and understandable. The parallel drawn from the Novak and Hochegger bistability paper needs to be better explained, and also, the seeming controversy of plk1 as a potential activator of Cdc25 and Cdk1: why it delays the Cdk1-CycB activity in unperturbed cycle while causes premature and progressive Cdk1 activity accumulation during the prolonged prophase in the inhibition experiment? After these concerns are addressed, I would be very enthusiastic to recommend the manuscript for publishing.

We would like to thank the reviewer for the positive evaluation of our work and very glad to see our shared excitement about mitotic entry regulation. In the revised version of our manuscript, we integrated the model originally published by Novák and Hochegger (Rata et al. 2018). This greatly helped to clarify concepts and indeed provided new insights and a much better understanding of the effects caused by plk1 inhibition. We would like to specifically thank the reviewer for this excellent suggestion.

In the published model, the initial trigger is provided by an increase in cdk2-cyclin A activity. Cdk2-cyclin A then activates the network through three parallel paths (see wiring diagram on new Fig. EV1E). Supported by the literature and our new data (Fig. EV1F), plk1 is involved in one of these paths, the path activating cdc25C. Thus, we incorporated plk1 in the branch downstream of cdk2-cyclin A and upstream of cdc25C (see Materials and Methods for details). We then simulated scenarios at 0%, 50%, 90% and 100% plk1 inhibition to recapitulate our dose-response experiments. However, the published model uses parameters that do not match the timescale of our experimental system, and parameters are chosen such that cdk1-cyclin B activity is near zero up until the feedback mechanisms kick in. Therefore, we scaled the parameters to match the timescale of our experimental system (new Fig. 1E), and adjusted the parameters so that phosphorylation of cdk1 substrates starts to slowly increase before the switch-like, full activation of cdk1 (new Fig. 2E, see Materials and Methods for details). This adjustment makes the model more realistic, since it is widely accepted that cdk1 activity is present already in prophase, before the full activation at NEBD, and we directly visualize this early increase in cdk1 substrate phosphorylation by the cdk FRET probe. These adjustments did not affect the overall behavior of the network and simulations robustly recapitulated key experimental observations in a broad range of parameters.

Plk1 inhibition caused a delay in mitotic entry without affecting the overall behavior of the network that showed bistability in all scenarios. However, plk1 inhibition substantially slowed the feedback mechanism, and therefore delayed the switch-like, full activation of cdk1-cyclin B and subsequent NEBD -- exactly matching experimental observations (new Fig. 1E and Fig. 2F). We further found that the extent of delay induced by plk1 inhibition is highly sensitive to variabilities in certain parameters. We chose to make small changes to the initial cdk2-cyclin A activity, which is a biologically meaningful source of variability. Small variations in initial cdk2-cyclin A activity could indeed explain the large cell-to-cell variability in the length of the delay observed in experiments (new Fig. 1E). Importantly, simulations also closely recapitulate the experimentally measured gradually increasing phosphorylation of cdk1 substrates both in unperturbed and in plk1 inhibited case; during the delay caused by plk1 inhibition, cdk1 substrate phosphorylation increases slowly and gradually (new Fig. 2E). Simulations recapitulate experimental observations that in the plk1-inhibited case cdk1 substrate phosphorylation reaches substantially higher levels before the positive feedbacks are triggered. This results from the slowed feedback mechanisms that require a higher cdk1 activity to sustain positive feedback.

Taken together, with minimal adjustments to the originally published model, simulations capture key features of plk1 inhibition observed in experiments: the dose-dependent delay of mitotic entry, the highly variable

entry times, as well as cdk1 activation kinetics. Thereby, the model provides an explanation for many if not all of the concerns of the reviewer. We integrated these results in the manuscript, which required relatively major changes to the text throughout, but we believe that this resulted in a substantial improvement in terms of conceptual clarity.

1. The study demonstrates that upon plk1 inhibition the cdk1 activity increases gradually over several hours with individual cells stochastically reaching the mitotic entry threshold. Based on this and other experimental observations, the authors conclude that plk1 acts as an effective catalyst of prometaphase transition, ensuring that NEBD occurs robustly and within a narrow time window. However, they also mention one major pathway (Gheghiani et al., 2017) identified to activate cdc25 as follows: cdk1/2-cyclin A phosphorylates the protein Bora, leading to the activation of Aurora A, which in turn phosphorylates and activates plk1. Plk1 then phosphorylates and activates cdc25C, which then dephosphorylates and activates cdk1. The authors should explain how inhibiting the plk1 activating function towards cdk1 will, in fact, prematurely activate Cdk1 during the prolonged prophase, albeit in a slower and progressively accumulating mode, but still, the plk1 inhibition seemingly activates cdk1. Could there be another branch of plk1 pathway in prophase that delays the cdk1 activity? Could the authors predict it from the proteomics data?

As explained above, simulations based on the model published by Novák and Hocegger have been extremely helpful to clarify this point. Based on the work of Gheghiani et al. we incorporated plk1 in the model as an activator of cdc25C downstream of cdk2-cyclin A. Consistently, plk1 inhibition slows the accumulation of active cdc25 and results in a delayed positive feedback and thereby delayed mitotic entry (note however that plk1 inhibition does not prevent mitotic entry, as cdk2-cyclin A activates the network through parallel paths, too) (new Fig. 1E). In addition, simulations also capture the experimental observation that cdk1 substrate phosphorylation continues to slowly increase during the delay caused by plk1-inhibition, reaching levels substantially higher than in unperturbed prophase (new Fig. 2E). In the absence of plk1 the level of active cdc25 increases slowly and a higher cdk1 activity is required to sustain the cdc25-cdk1 positive feedback that in turn drives the switch-like activation of cdk1. This is an important finding that is consistent with our experimental data: the cdk1 FRET probe, cyclin B localization as well as cellular phenotypes all point to the fact that cdk1 activity reaches higher levels during prolonged prophase as compared to unperturbed prophase.

2. A scheme, or a wiring diagram, analogous to the one presented in Novak & Hocegger paper (Figure 3) would help the general reader a lot. It could explain how the authors envision the regulatory connections among the regulators plk1, cdk1, wee1, cdc25, mitotic substrate pool, etc.

We now include a wiring diagram as well as a detailed description of the simulations we performed (Fig. EV1E and Materials and Methods).

3. Similar concerns as in Point 1 above apply to the following interpretation: "In normal prophase, these early changes to the nuclear envelope have been proposed to facilitate nuclear accumulation of cdk1-cyclin B (Gavet and Pines, 2010), which is then key to full activation of cdk1, driving NEBD and transition to prometaphase (Santos et al., 2012). Plk1 inhibition may prevent these changes in nuclear transport causing the observed slow and limited relocalization of cyclin B to the nucleus and thus slow and gradual increase in cdk1 activity." How does the plk1 inhibition in Figure 4 cause both accumulation of Cyclin B and cdk1 activity: "cdk activity detected by the FRET probe was almost identically mirrored by slow and gradual relocalization of cyclin B1 to the nucleus?"

As explained above, simulations based on the model by Novák and Hocegger were instrumental to explain this apparent controversy. The simulations reveal that plk1 is required for an efficient and fast activation of the positive feedback leading to the switch-like full activation of cdk1-cyclin B. Plk1 inhibition slows the activation of cdc25 that is required to initiate the cdc25-cdk1 feedback, that in turn causes a delay in complete activation of cdk1 required for NEBD. During this time, active cdk1 is not zero but continuously increases until it is sufficiently high to sustain the positive feedbacks. This leads to an accumulation of cdk1 substrate phosphorylation. This result of the computer simulations matches experimental observations: the cdk1 FRET probe, cyclin B localization and cytoskeletal behavior all indicate that cdk1 substrate phosphorylation reaches higher levels during prolonged prophase as compared to unperturbed prophase.

4. How do the authors interpret the relationship between the slow accumulation of cdk1 activity (plk1 inhibition) vs abrupt switch (no inhibition) and the progression of phosphorylation dependent events in prophase? It is unclear how these rare "stochastic" mitotic entries in plk1 inhibitor experiment can occur in

case of slow cdk1 accumulation, which would likely mean that only an occasional incremental change can trigger the mitosis in these stochastic events, while in normal cell cycle one needs a switchlike input of cdk1.

Again, incorporating the model and performing simulations clarified these points. Firstly, as detailed above, the simulations explain both the delay caused by plk1 inhibition and the slightly increased cdk1 substrate phosphorylation accumulating during this extended prophase-like state. Without the model, we interpreted our observations as stochastic events, but after incorporating the model in our manuscript, we realized that this may not be the most likely explanation. As shown on the new panels Fig. 1E, a more feasible and biologically meaningful explanation for the highly variable entry times are small variabilities in the initial cdk2-cyclin A levels. This may stem from biological variability between individual cells, but even more simply, it can also result from slight asynchrony within the cell population.

Also, as the cyclin expression levels are very variable and overlapping when populations of individual cells are analyzed (Figure 4 E,F), how can the tight and abrupt switch be explained by a cdk1 threshold? These thresholds must be really flexible?

The error bars are indeed relatively large, but the primary source of this is the imaging assay and automated image analysis -- cells frequently move in and out of the focus, which leads to fluctuations in fluorescent intensities. As seen on the individual cell traces (Fig. EV2A, B), there is relatively little variability in the overall shape of the traces, it is rather the 'bumpiness' caused by focus shifts that degrades the quality of the data. Thus, there is no need to assume variable thresholds.

5. In connection with the previous, and the model recently published by the Hochegger and Novak groups (Rata et al., 2018) suggesting that mitotic entry relies on two inter-linked bistable switches, the positive feedback loop activating cdk1, and the feedback through Greatwall leading to the inactivation of the PP2A:B55 phosphatase. However, Hochegger and Novak did not integrate the plk1 loop into their model and wiring diagram. Please explain how the plk1 enters into the diagram, and most importantly, how the non-stable steady state that was predicted by modeling and experimentally demonstrated in Hochegger and Novak paper by double inhibition of Wee1 and Greatwall is analogous, if not the same intermediate state, that the authors show in the current paper, as a result of plk1 inhibition.

We have incorporated plk1 into the model downstream of cdk2-cyclin A and upstream of cdc25C based on the experimental data by Gheghiani et al. (Fig. EV1E). In light of the simulations and analysis of the results, we needed to reconsider our previous discussion. In the model, cdk2-cyclin A activates cdk1 by acting on three pathways: wee1, Greatwall and cdc25, of which plk1 is only involved in the pathway leading to cdc25c activation. Therefore, plk1 inhibition merely slows down the activation of cdc25, but otherwise it does not seem to change the behavior of the network, and bistability is maintained from 0-100% plk1 inhibition (Fig. EV1G). Plk1 inhibition does not appear to establish a new steady state, and thereby the effect of plk1 inhibition is distinct from the 'hidden steady state' observed by Rata et al. by double inhibition of wee1 and Greatwall. We have therefore rewritten the corresponding paragraph of the Discussion.

6. If the activation is cdc25 phosphorylation is dependent on plk1, then accumulation of Cyclin B and inhibitory phospho-Tyr on cdk1 should not be in correlation (when plk1 is inhibited). Or at least, higher plk1 inhibition doses should cause more phospho-Tyrosine signal. Can this be tested quantitatively, similarly as in Fig 4 D and E in the Hochegger paper?

We performed Western blots using a phospho-cdc25C antibody and the cdk1 phospho-Tyr antibody, now shown in Fig. EV1F. These blots reveal, firstly, that in plk1 inhibited cells cdc25C phosphorylation is completely absent at all stages (early and late in prolonged prophase as well as in the *polo* prometaphase arrest stage). The cdk1 phospho-Tyr signal is maintained at high levels until late in prolonged prophase, but it is, as expected, absent in the *polo* prometaphase arrest. These observations confirm previous findings by Gheghiani et al. and are generally consistent with our model and other experimental data.

7. The authors could perhaps tone down the following statement: "Prolonged prophase thus serves as a model for an otherwise inaccessible transient stage of mitosis, early, before cdk1 is fully activated." It is still a very much different situation as the inhibitor treatment changes a lot of signaling dynamics or likely hundreds of kinase targets compared to the normal prophase. "Thus, plk1 inhibition may interfere with the positive feedback loop activating cdk1, while the PP2A:B55 branch remains intact. In this context, prolonged prophase may be an ideal condition to analyze the complex signaling network regulating mitotic entry." Similar concern: in case cdk1 activity slowly increases and the phosphatase is not affected, then the phosphorylation stoichiometries hardly resemble prophase situation and cannot be called as an "ideal condition".

Here again, incorporating the computational model into the manuscript helped tremendously to clarify these points. Therefore, we have extensively rewritten the corresponding paragraphs. In essence, we agree with the reviewer that the prolonged prophase-like state in plk1-inhibited cells differs in a couple aspects from normal prophase: plk1 activity is absent, and as we show, cdk1 substrate phosphorylation reaches levels higher than in normal prophase. On the other hand, this state does resemble prophase in several critical aspects: it is a state early in cell division with low cdk1 activity (before full activation of cdk1) and with nuclear envelope still intact. Therefore, our phosphoproteomics data does provide a unique insight into early events of cell division that regulate mitotic entry. We have reformulated our Discussion accordingly.

Referee #2

This manuscript by Gobran and colleagues reports the effects of PLK1 inhibition on the timing of mitotic entry and mitotic progression. These effects have been studied before, including in a seminal 2007 paper from the manuscript's senior author. In this new study, the authors used specific PLK1 inhibitors to verify two fundamental reported effects of PLK1: 1) a (usually time-limited) arrest in mitotic prophase and 2) an arrest in prometaphase with a "polo" phenotype (essentially a monopolar spindle). After characterizing the prophase arrest in detail, the authors came to the conclusion that it represents a special state, at least in part distinct from G2, and performed a phosphoproteomic analysis to identify phosphorylation sites characteristic of this phase.

Briefly, in Figure 1 the authors use three distinct human cell lines to determine that PLK1 inhibition (with the specific and selective inhibitor BI2536) delays mitotic entry and, after mitotic entry, causes cells to display a polo phenotype. In Figure 2 they demonstrate premature chromosome condensation long before mitotic entry. In Figure 3, they demonstrate that several aspects of cell physiology, including microtubule and actin dynamics, but not nuclear transport, are perturbed during a prolonged prophase. In Figure 4, they use a CDK FRET sensor to investigate the timing of CDK mitotic activation, and demonstrate that CDK activity grows progressively in PLK1-inhibited cells, rather than abruptly as in control cells. In Figures 5 and 6 the authors present the results of a phosphoproteomics analysis that identified various phosphorylation sites in G2, prophase-like, and prometaphase states. An important conclusion of this analysis is that phosphorylation status in the prolonged-prophase state induced by PLK1 inhibition is more similar to that in G2 cells than to that in prometaphase cells, although differences were also measured.

Overall, the study is rigorous and technically very well done, and the controls are adequate. The conclusions consolidate and extend previous, more scattered evidence. The proteomic analysis provides an important new tool for researchers interested in the mechanisms of cell cycle control. On the weak side 1) there is a significant conceptual concern, as explained below; 2) the study remains somewhat superficial on the mechanistic level, although this may be expected given that this is an already significant contribution. In this context, a more effective presentation of the results of the proteomics analysis would give the manuscript a more mechanistic touch that is currently missing.

We would like to thank the reviewer for the overall positive evaluation of our work!

Specific points

- Prophase is intrinsically transient. The authors may have identified a condition to prolong prophase, but this condition is achieved by eliminating PLK1, which is a crucial component of the switch and itself also a kinase targeting many different substrates. This is clearly different from activating a checkpoint to block cells in a certain cell cycle regime without touching the relevant cell cycle kinases (of note, there is no prophase checkpoint that we know of). Chromosome condensation and the dotted pH3 S10 signal are the only hallmark of prophase observed by the authors. Other observed aspects, e.g. microtubule plus-tip velocities, do not fit the G2 or the M-phase pattern, being even higher than in mitosis. Thus, I am doubtful that the state studied by the authors is, strictly speaking, a prolonged prophase, at least not in the way that we consider a prolonged mitosis the arrest obtained with a spindle poison, for instance. A possible way to address this conundrum would be to extend the proteomics analysis to include a comparison with cells arrested before mitosis through inhibition of CDK1 with RO3306, a condition that the authors, in the Discussion, equate to prophase.

We would like to thank the reviewer for pointing us to this important conceptual distinction. We now modified the manuscript text at several places to clarify two important points:

Firstly, the reviewer is right that it is very important to distinguish slowed progression from a checkpoint-type arrest. Our data using cell cycle markers, the cdk1 FRET probe in particular, clearly evidence that in absence of plk1 activity prophase progression is not blocked, it is just slowed and cdk1 substrate phosphorylation is increasing slowly and gradually over hours. To use a trivial analogy, when driving a car, a checkpoint arrest would correspond to being halted at a red light, while slowed progression would correspond to driving in first gear -- while both will cause the driver to spend longer time on the road, the reasons are different. We still think that 'prolonged prophase' is an appropriate term to describe the state caused by plk1 inhibition, but we clarified and slightly toned down our statements accordingly.

Secondly, following the suggestion of reviewer 1, we incorporated a previously published computational model of the mitotic entry network in our manuscript. We believe that this addition greatly helped to clarify the mechanisms by which plk1 regulates prophase progression. To the reviewer's questions here, simulations clarified two important points:

- The model clarifies that plk1 slows down the positive feedback mechanism of cdk1 activation causing an entry delay. Cdk1 inhibition is expected to have a different effect: the feedback mechanisms are intact in this case, and therefore the network will respond in a switch-like manner. As a consequence, the response to RO-3306 is expected to be binary; cells either remain in G2 or enter mitosis. Indeed, we attempted this experiment and we could confirm the behavior as predicted by the model (data now shown). We were unable to find an intermediate concentration of RO-3306 that would reduce cdk1 activity so that we capture cells in a prolonged prophase-like state. We are currently working on alternative synchronization protocols to simultaneously disable some of the feedback mechanisms combined with RO-3306 to enable a gradual tuning of cdk1 activity, however these works are beyond the scope of the current manuscript.
- Secondly, we simulated cdk1 substrate phosphorylation during prophase in control and plk1-inhibited cells. Remarkably, these simulations recapitulated the continuous increase in cdk1 substrate phosphorylation during prolonged prophase reaching levels substantially higher than in control prophase cells. The reason for this is that plk1 inhibition slows the accumulation of active cdc25 and so the capability to sustain the cdc25-cdk1 positive feedback. As a consequence, a slightly higher level of cdk1 activity is required to initiate and sustain the positive feedback (new Fig. 2E, F). This new finding also explains why cytoskeletal changes and chromosome condensation reach higher levels in plk1-inhibited cells than in unperturbed prophase. Therefore, on the one hand, we agree with the reviewer that we should be cautious in stating to what extent our prolonged prophase is similar to unperturbed prophase. On the other hand, our data and simulations together define this as a unique state with low cdk1 activity and an intact nucleus, preceding the full activation of cdk1 and NEBD. Thereby, this state recapitulates critical aspects of prophase and provides insight into the events that occur early in cell division.

-From section 2 of the results onwards, the authors perform all their experiments in HeLa cells. It would be useful to state so in the main text, and clarify this choice for the readership.

We now state this clearly in the text.

-Figure 4C: CDK1 FRET signal is less steep but higher in the BI condition. Can the authors comment on this?

As also mentioned above, we consistently observed a slow and gradual increase in cdk1 substrate phosphorylation in plk1-inhibited cells that reaches levels substantially higher than in unperturbed prophase (Fig. 2C). We observed a similar trend in nuclear accumulation of cyclin B (Fig. 2D), additionally cytoskeletal changes and chromosome condensation also consistently show stronger effects in plk1 inhibited cells than in control prophase (Figs. 3 and 4). Having included the computational model in our manuscript, we are now able to interpret these observations (see above).

-Figure 5A: Why not compare BI+STLC at 5h and 13.5h? Wouldn't STLC at 5h, BI+ STLC at 5h, STLC at 13.5h, BI + STLC at 13.5h have been a better control?

Based on our live-imaging experiments (Fig. 1C, EV1A) and the cdk kinetics measured by the FRET probe (Fig. 2), in the BI+STLC at 5h sample we would expect many of the cells to be still in G2 with very low cdk1 activity.

Therefore, in these cells we would not expect to see the phosphosites typical for the prolonged prophase state just yet.

-The important proteomics data in figures 5 and 6 is presented rather sketchily. Readers can only access the critical information by referring to the tables, which is time-consuming. Which enriched or depleted phosphosites in the prolonged prophase group are actually specific prophase phosphorylations? Are the identified sites merely more or less abundant "copies" of G2 or prometaphase sites? Figure S4B is nice and could be placed in the main figures, extended to include the prophase enriched sites. This would be more informative, as currently the identified sites are classified based on the function of the phosphorylation protein, and any information on the progression from G2 to prophase to mitosis is missing. In addition to the GO classification, it would be important to annotate which of the identified proteins are known to be direct PLK1 substrates (based on previous literature) and whether the identified sites conform to a consensus. Also, how many sites are in common between G2 and the ProPro state, how many between ProPro and M, etc.? This could be easily presented with the classical "overlapping circle" plots.

Following the reviewer's advice, we attempted to improve the visualization of our proteomics data in order to provide more details. Firstly, we mapped a network of the significant hits using the STRING database (Appendix S1). This allowed us to find clusters of proteins that are significantly phosphorylated in the prolonged prophase state. One of the largest clusters of 44 nodes is enriched for the GO terms 'regulation of chromosome segregation' and 'cell cycle checkpoints', and is now shown on Fig. 6A. We also made adjustments to Fig. 5C. We now show all the four pairwise comparisons between the four samples and we also included in the main text a comprehensive explanation of our fifth comparison, which shows hits specific to Prolonged Prophase. Also, we now highlight in figure EV5D all the cyclin dependent kinases that were shown to be enriched in ProPro. This analysis confirms that BI 2536 primarily affects plk1 substrates, whereas in the prolonged prophase state most sites are cdk1 phosphorylations.

Minor points

-I would recommend removing adverbs of subjective surprise and interest, from "strikingly" to "intriguingly". The last word on this is with the editors, though.

We removed these adverbs in the majority of cases -- we apologize for our overexcitement.

-Introduction, end of first paragraph: the authors should clarify that it is possible to arrest cells in G2 with RO3306, and explain in what way this is different from what they want to do. This is all the more important in view that in the Discussion the authors claim that the prolonged prophase arrest they have obtained by inhibiting PLK1 phenocopies CDK1 inhibition.

Adopting the computational model originally published by Rata et al. and performing the simulations clarified that the parallel we have drawn is not fully justified. On the one hand, as detailed above, we could easily incorporate plk1 into the published model and simulations readily recapitulated key experimental observations, and thereby it greatly helped in clarifying and strengthening our conclusions. On the other hand, it also became clear that the effect of plk1 inhibition is distinct from those observed by Rata et. al. Their main finding is that simultaneous inhibition of wee1 and Greatwall can drive cells in a normally hidden steady state. Our data and simulations show that plk1 inhibition does not drive cells into a steady state, it just slows down a transition. On the experimental side, Rata et. al. uses a cell line expressing an ATP-analog sensitive cdk1 mutant. Taking advantage of this system, they use an advanced protocol whereby release from 1NM-PP1 followed by re-addition of increasing concentrations of 1NM-PP1, which is very different from treating cells in G2 either with the plk1 inhibitor or RO-3306 (which is likely to partially affect cdk2 as well).

Taken together, we now show that effects of plk1 inhibition and complex treatments performed by Rata et al. can be well explained by the same universal model of the signaling network. The effects are distinct, but it would certainly be interesting to combine some of these treatments in the future.

-Results, first paragraph: "...compare side-by-side three cell lines..."

We changed the phrase as suggested.

-Results, second paragraph: "...we escalated doses to multiples of the concentrations shown to suffice..."

We changed the phrase as suggested.

-The authors show that S-phase progression is not affected by PLK1 inhibition (Figure S1A-C). The authors may consider including a marker for G2-M progression to show that also G2 is not affected, and specifically by monitoring the disappearance of PCNA speckles.

In figure EV1A-C we show the time duration from S-phase termination (marked by the disappearance of PCNA speckles) until NEBD, that is the combined duration of G2 and prophase. We show that this duration is increased when plk1 is inhibited.

-Figure 3D would be more intuitive to read if the authors explained directly in the main text what the magenta arrow means.

We now included this explanation.

-Page 5, last paragraph: "regulatory kinases"; which ones?

We changed the text accordingly.

-Throughout: "data" is plural, i.e. data are, not is

We changed 'data' to plural throughout the manuscript as requested by the reviewer. (Of note, while in the latin origin of the word 'data' is the plural form of 'datum', according to most dictionaries the use of 'data' with a singular verb is broadly accepted and entirely standard in both American and British English. In this case it is being treated as a noncount noun, like 'information'.)

-Page 7, third paragraph: "...harboring phosphosites of different abundance"

We changed the text accordingly.

-Page 8, elimination of CENP-F has absolutely mild effects on kinetochore assembly and the paper referred to demonstrated a role in Dynein-Dynactin recruitment or activation at the kinetochore.

We changed the text and exchanged the reference to focus on the role of CENP-F in stabilizing microtubule-kinetochore attachments.

-Figures 5 and 6 (including legend): what does "specific to prolonged prophase" mean exactly? What is being shown exactly in the volcano plot in Figure 5? Is it a comparison with G2? I could not tell what exactly is displayed.

We used linear modeling to analyze our data, a standard approach to analyze datasets that change along not only one variable, but multiple variables. Specifically, in our analysis one variable or 'axis' is plk1 inhibition (+/- BI 2536) and the other axis is mitotic time (before or after NEBD) (Fig. 5C). We then identified the phosphosites that are specific to prolonged prophase, i.e. to plk1 inhibition AND before NEBD. This does not include the phosphosites that are common to plk1 inhibited samples (i.e. BI 2536 treated before and after NEBD). It also excludes the sites that are common to samples before NEBD. We now explain how the analysis was done in more detail in the manuscript text.

-Figure 2E: Two different scales are used, but it seems the percentages of dotted cells after release are quite different in the control and BI-treated samples. Any hint why that is the case?

The reason is that control cells spend only a short time in prophase, and undergo NEBD in 5-10 minutes after these 'dotted cells' start to appear. Due to the imperfect synchrony of cells, at any given time there is only a relatively small proportion of the cells in this transient state. In BI2536 treated cells, as prophase lasts several hours, the 'dotted cells' accumulate over time, and at later time points nearly all non-rounded-up cells are in the prolonged prophase state -- a critical advantage enabling our phosphoproteomics experiment.

-Discussion, first paragraph: "somewhat contentious". Relevant references have to be added to support the statement.

We added the reference.

-In Figure S2, Nup107 signal looks less clear of LBR and it seems more intense on the chromosomes at NEBD in the BI condition. Can this signal be quantified as well? If not, it may be better to remove it.

We removed the figure as suggested.

- Figure S3C should be called as S3A as it comes first in the main text.

We did a major rearrangement of the figures

-It would be nice to include the number of cells in each graph where the population average behavior was plotted.

We changed the graphs accordingly.

-The authors should consider including statistical analysis to Figure 3B.

We changed the figure accordingly.

-In the beginning of section 4, the opening sentence "Thus, prolonged prophase ... and microtubule cytoskeleton" sounds like the conclusion of a paragraph rather than the beginning of a new one. I would suggest rephrasing it for clarity.

We do think this sentence helps the flow of the manuscript, and therefore we left it unchanged.

-The authors could spend more time discussing interesting potential targets. For instance, do they detect differential phosphorylation by the Cdc25C1 phosphatase in their prolonged prophase sample? This would make their point stronger.

For unknown reasons, we failed to detect phosphopeptides of cdc25C in our phosphoproteomics experiment. However, we performed Western blots with a phospho-cdc25C antibody and could show that cdc25C phosphorylation is prevented completely when plk1 is inhibited, confirming previous findings by others.

-In section 6 of the results, the authors show that NCAPH2 and NCAPD3 are differentially phosphorylated in the prolonged prophase sample. Since it is now known that M18BP1 phosphorylation by CDK1 recruits Condensin II to chromosomes during mitotic entry, it would be nice to also show whether M18BP1 is detected in the screen.

M18BP1 was detected in our phosphoproteomics experiment. Its phosphorylation state is not significantly changed in prolonged prophase, but when comparing mitotic and interphasic samples, we do detect a significant change in phosphorylation on T149. This and all other changes we detected in our proteomics experiment are listed in Table S3.

Referee #3

Plk1 is a key cell cycle kinase which has important roles in regulating mitosis at different stages. One critical role for Plk1 is the control of mitotic entry, and it had already been observed 20 years ago that RNAi-mediated depletion of Plk1 delays mitotic entry, an observation that was subsequently confirmed by small molecule inhibition of Plk1 (Sumara et al., 2004; Lenart et al., 2007). The effect of delaying entry into mitosis and prolonging prophase is most likely due decreased activating phosphorylation of the CDC25C phosphatase by Plk1, resulting in delayed removal of the inhibitory Thr14/Tyr15 phosphorylations, and it has been previously shown that phospho-mimetic versions of CDC25C largely rescue the mitotic entry defect in cells treated with Plk1 inhibitor (Gheghiani et al., 2017).

In this manuscript Lenart and colleagues use live cell imaging to carefully analyse the mitotic entry delay in different cell lines, and characterise the effects on chromosome condensation, the microtubule cytoskeleton and cyclin B localisation. They come to the conclusion that Plk1 inhibition results in a prolonged prophase, and then aim to exploit this observation by performing a mass spec analysis of these cells in comparison with mock-treated cells to identify phosphorylations that characterise prophase.

This is a nicely conducted manuscript with high-quality data. Unfortunately, though, the study is almost entirely descriptive and provides very little novel insight into either the role of Plk1 in regulating mitotic entry or the biology of prophase. Surprisingly, the supposedly key effector of Plk1, the phosphatase CDC25C is not analysed in the manuscript. It would be informative to know whether the phosphorylation status of CDC25C and its substrate Cdk1 change in accordance with expectations and whether all the changes that Gobran et al. observe are due to insufficient CDC25C activation. Furthermore, it would be interesting to know whether the Plk1 inhibition phenotypes could be rescued by expressing a phospho-mimetic version of CDC25C, although this line of experimentation is very close to the experiments that have already been published by Gheghiani et al., 2017.

We thank the reviewer for appreciating our efforts and the quality of our experimental data. We also agree with the reviewer that our data builds on and is fully consistent with previous studies. However, we disagree in that our manuscript would not provide any novel insights into plk1's function during mitotic entry:

1. Despite those long published studies referred to by the reviewer, a controversy persisted whether plk1 inhibition delays or completely prevents mitotic entry. We performed carefully controlled dose-response experiments in three different cell lines, and, as suggested by reviewer 1, we now include a

mathematical model to explain these observations across cell lines. The model fully resolves the controversy revealing that the apparent variability of the phenotype is an inherent property of the signaling network, and is not a consequence of experimental issues as presumed previously.

2. We show that plk1 inhibited cells are delayed in a prophase-like state with slowly increasing cdk1 activity, displaying several hallmarks of prophase, including chromosome condensation, increased microtubule dynamics and show early signs of rounding up. This has not been shown before.
3. Our phosphoproteomics survey provides for the first time a catalog of phosphorylation events taking place in a prophase-like state early in mitosis when cdk1 is not yet fully activated and the nuclear envelope is still intact. This survey reveals a specific set of phosphosites, much fewer and very distinct from the sites phosphorylated in prometaphase when cdk1 is fully activated. The functional annotation of the hits also matches closely the cellular effects observed on nuclear organization and cytoskeletal dynamics.

Regarding the role of cdc25C, we now performed Western blots to show that cdc25C is not phosphorylated in plk1 inhibited cells, and as expected, cdk1 is activated with a delay as judged by Tyr15 phosphorylation. The rescue experiment using the phospho-mimetic cdc25C mutant we did not perform, as we think this has already been conclusively shown and carefully documented by Gheghiani and coworkers.

The proteomic analysis of the prolonged prophase state found in plk1-inhibitor treated cells, could in principle be interesting but in its current form, although the analysis of the data identifies proteins that are differentially phosphorylated in the Plk1-inhibitor-treated cells, there is no identification of the kinases that normally carries out these phosphorylations. This should most likely be Cdk1-cyclin B1 but it would be good to have this confirmed.

We performed kinase enrichment analysis in order to identify the kinases most likely to be responsible for the detected phosphorylations. This reveals that cyclin-dependent kinases (cdks) are primarily responsible for prolonged-prophase-specific phosphorylations (Fig. EV5D) (Note that this bioinformatic tool has somewhat limited sensitivity, so it is unlikely to be able to distinguish between cdks.)

Can the authors identify any key substrates that would push cells into mitosis if a phospho-mimetic mutant was expressed?

Most proteins we identified by phosphoproteomics are downstream effectors, regulators of chromatin organization, the nuclear envelope or cytoskeletal dynamics. We do not expect that phospho-mimetic mutants of these downstream effectors would have a substantial effect on the signaling network controlling mitotic entry. The mathematical model, now incorporated in the manuscript, makes clear predictions for how the signalling network can be altered to artificially drive mitotic entry, some of which are tested in that study. For example, it is predicted that wee1 inhibition advances mitotic entry independent of the plk1-cdc25C branch of cdk1 regulation (see wiring diagram on Fig. EV1E). We tested this by using the small molecule compound AZD1775, a specific inhibitor of wee1. AZD1775 treatment opposed Plk1 inhibition by BI2536, reversing the delay in mitotic entry. We feel that this data does not fit the present flow of the manuscript, and therefore we included the figure below for the reviewers' discretion.

Altogether, this is a nice confirmation of the role of Plk1 in promoting mitotic entry but there is little advance over already published studies.

See our comments above pointing to the novel findings in our study, not shown in previously published work.

Minor comments:

-Table S3 with the list of differentially phosphorylated proteins in prophase was not included with the submitted manuscript, hence an evaluation of these data was not possible.

We now include these data both to the figures and as a separate table.

-"We also identified phosphorylation of the TTK/MPS1 kinase, which regulates the assembly of the mitotic checkpoint complex on the kinetochore (Schweizer et al., 2013)."

"Schweizer et al., 2013" is not the correct reference for this statement. Mitotic phosphorylation of human MPS1 has been demonstrated by Diril et al., 2016, Plos Genet.; and Hayward et al., JCB, 2019, and these two publications would be better references for this statement.

Thank you for pointing this out, we changed the references as suggested.

Dear Peter,

Thank you again for submitting your revised manuscript (EMBOJ-2024-118013R) to The EMBO Journal for our consideration. As I have already informed you, it has been seen by the three referees who had previously also assessed the original version of your manuscript, and I am glad to say that they are all satisfied with the revision, mentioning that almost all initially raised concerns have been successfully addressed and the manuscript significantly improved. There are only three minor points from referee #1 requiring further clarification/correction in a final version of your manuscript, before we can proceed with its publication in The EMBO Journal. Please see the referees' comments below.

There are also a few editorial requests/formatting changes that we need from you to address in the final version of your manuscript:

- Please provide a list of up to 5 relevant keywords after the Abstract of your revised manuscript.
- We kindly request you to deposit all mass spectrometry datasets produced in this study in appropriate databases, and to provide the database, identifiers, and specific URLs to the datasets in the Data availability section of your manuscript (suggested wording: "The [structural coordinates | microarray | mass spectrometry] data from this publication have been deposited to the [name of the database] database [URL] and assigned the identifier [accession | permalink | hashtag]."). Please make sure to include the permanent links to the deposited datasets.
- A conflict-of-interest statement under the heading "Disclosure and competing interests statement" is mandatory. Please review our journal's policy here (<https://www.embopress.org/page/journal/14602075/authorguide#conflictsofinterest>) and include this statement in the final version of your manuscript.
- The author contributions statement should be removed from the manuscript file. Instead, we use CRediT to specify the contributions of each author in the journal submission system. Please feel free to use the free text box to provide more detailed descriptions during submission. See also our guide to authors for more information: <https://www.embopress.org/page/journal/14602075/authorguide#authorshipguidelines>.
- As per our journal's policy, "data not shown" (on page 19) is not permitted. All data referred to in the paper should be displayed in the main or Expanded View figures, or in the Appendix. Please add these data or change the text accordingly if these data are not central to the study and its conclusions, or properly cite the respective published sources if these data can be found elsewhere.
- We noticed that the third column with manuscript sections has not been completed in the section "Data Availability" of your Author Checklist. Could you please add this information and upload the revised checklist along with your re-submission?
- Please note that the Expanded View (EV) Figures should be uploaded as individual, high-resolution Figure files, while their legends should remain in the main manuscript file.
- The Dataset legends (labeled as "Supplemental Table S1-S4") should be removed from the manuscript file and included in each corresponding Excel file as a separate tab/sheet, or above the table in .txt files. For Tables S1-S3, please update all source file names, titles, legends, and manuscript callouts to "Dataset EV1-EV3". Table S4 should be renamed to "Table EV1" (also update source file name, title, legend, and manuscript callouts) with the legend included above the table in the .doc file and uploaded as an Expanded View Content (was "Supplementary Information") file.
- The first page of your Appendix pdf file should include the heading "Appendix" and the title of your manuscript, followed by a brief Table of Contents including the page numbers of the listed items. The nomenclature of the Appendix items should be "Appendix Figure S#" and "Appendix Table S#" throughout the Appendix pdf file and the corresponding callouts in the main manuscript file. The Appendix Figure legends should be removed from the manuscript file and included in the Appendix pdf file below the respective Figures.
- Please note that the Reagents and Tools Table should be removed from the manuscript file and instead be uploaded as an individual file using the template and instructions in our guide to authors: <https://www.embopress.org/page/journal/14602075/authorguide#structuredmethods>. Please upload your completed Reagents and Tools Table to our manuscript tracking system as a "Reagent Table" file.
- Please specify in your Source Data checklist whether the Source Data for Figure 5E have been deposited in an external repository as suggested by our Source Data coordinator.
- Please note that EMBO press papers are accompanied online by:
A) a short (2 sentences) summary of the findings and their significance,

B) 2-5 short bullet points highlighting the key results, and

C) a synopsis image in .jpg or .png format that is exactly 550 pixels wide and 300-600 pixels high (the height is variable). Please note that the text needs to be legible at the final size.

Please upload this information along with your revised manuscript (the text for A and B should be provided in a separate Word file).

- During our routine pre-acceptance checks, our data editors have raised the following queries regarding figures, data, and legends. Please make sure that all requests below are completely addressed in the final version of your manuscript:

1. Please note that the exact p values must be provided in the legend of Figure EV1 C.
2. Please indicate the statistical test used for data analysis in the legends of Figures 5D, E; 6A-C; EV5 B-D.
3. Please indicate what */ **/ ***/ **** represents; if this represents p value(s), please indicate the statistical test used and where appropriate provide the exact p value in the legend of Figure EV2E.
4. Please note that the box plots need to be defined in terms of minima, maxima, centre, bounds of box and whiskers, and percentile in the legend of Figure EV5A.
5. Please note that information related to "n" is missing in the legends of Figures 2C, D; 5E; EV2 C, E.
6. Please note that the scale bar needs to be defined in the legend of Figure EV3 A.

- We also note that the section order in your manuscript should be corrected as follows: Title page - Abstract & Keywords - Introduction - Results - Discussion - Methods - Data Availability - Acknowledgements - Disclosure and Competing Interests Statement - References - Figure Legends - main Table(s) (if there are any) - Expanded View Figure Legends.

Please also note that as part of the EMBO publications' Transparent Editorial Process, The EMBO Journal publishes online a Peer Review File along with each accepted manuscript. This File will be published in conjunction with your paper and will include the referee reports, your point-by-point response and all pertinent correspondence relating to the manuscript. You can opt out of this by letting the editorial office know (contact@embojournal.org). If you do opt out, the Peer Review File link will point to the following statement: "No Peer Review File is available with this article, as the authors have chosen not to make the review process public in this case."

We look forward to seeing a final version of your manuscript as soon as possible. Please let us know if you have any questions and use this link to submit your revision: <https://emboj.msubmit.net/cgi-bin/main.plex>.

Best wishes,

Ioannis

Referee #1:

The authors have added the mathematicam model and simulations which improved the paper considerably. The sentence ' The cause for this is that in the absence of plk1, the level of active cdc25 increases slower, and therefore a higher cdk1 activity is required to sustain the cdc25-cdk1 positive feedback, which in turn drives the switch-like activation of cdk1.' is still difficult to understand. The slower increase of active cdc25 should shift the equilibrium towards inhibited cdk1 which in turn shifts the equilibrium towards active wee1. This does not cause the observed 'higher cdk1 activity'. Quite the opposite. It should be explained better. One sentence like this is not sufficient.

Minor points.

- 1) Figure EV1 panel E: the ENSA scheme is misleading. The same released free phosphatase seems to be specialised on different tasks via diferent arrows. Likely not true.
- 2) Under the equations there is KPPX,Y5, but in the Table S4 there is KPPX,Y15.

Referee #2:

In this revised version of the manuscript, the authors have greatly improved the presentation, and have included and revised a previously published computational model to describe the role of PLK1 and its inhibition on the G2/M transition. I am excited by the revised manuscript and strongly support its publication in EMBO J.

Referee #3:

The authors have carefully revised their manuscript and addressed all the concerns that have been raised. The manuscript has been significantly improved, and I am very happy to support publication.

All editorial and formatting issues were resolved by the authors.

Dear Peter,

Congratulations on an excellent manuscript! I am very pleased to inform you that it has been accepted for publication in The EMBO Journal. Thank you for comprehensively addressing the concerns initially raised by the referees, and all editorial and formatting requests.

If you have any questions, please do not hesitate to contact the Editorial Office. Thank you for your contribution to The EMBO Journal. Working with you has been a pleasure!

Best wishes,

Ioannis
